# Inference of the SARS-CoV-2 generation time using UK household data

**William S Hart[1]\***, **Sam Abbott[2]**, **Akira Endo[2]**, **Joel Hellewell[2]**, **Elizabeth Miller[3,4]**, **Nick Andrews[5]**, **Philip K Maini[1]**, **Sebastian Funk[2†]**, **Robin N Thompson[6,7†]**

[1]Mathematical Institute, University of Oxford, Oxford, United Kingdom; [2]Centre for the Mathematical Modelling of Infectious Diseases, London School of Hygiene and Tropical Medicine, London, United Kingdom; [3]Department of Infectious Disease Epidemiology, London School of Hygiene and Tropical Medicine, London, United Kingdom; [4]Immunisation and Countermeasures Division, UK Health Security Agency, London, United Kingdom; [5]Data and Analytical Sciences, UK Health Security Agency, London, United Kingdom; [6]Mathematics Institute, University of Warwick, Coventry, United Kingdom; [7]Zeeman Institute for Systems Biology and Infectious Disease Epidemiology Research, University of Warwick, Coventry, United Kingdom

**Abstract** The distribution of the generation time (the interval between individuals becoming infected and transmitting the virus) characterises changes in the transmission risk during SARS-CoV-2 infections. Inferring the generation time distribution is essential to plan and assess public health measures. We previously developed a mechanistic approach for estimating the generation time, which provided an improved fit to data from the early months of the COVID-19 pandemic (December 2019-March 2020) compared to existing models (Hart et al., 2021). However, few estimates of the generation time exist based on data from later in the pandemic. Here, using data from a household study conducted from March to November 2020 in the UK, we provide updated estimates of the generation time. We considered both a commonly used approach in which the transmission risk is assumed to be independent of when symptoms develop, and our mechanistic model in which transmission and symptoms are linked explicitly. Assuming independent transmission and symptoms, we estimated a mean generation time (4.2 days, 95% credible interval 3.3–5.3 days) similar to previous estimates from other countries, but with a higher standard deviation (4.9 days, 3.0–8.3 days). Using our mechanistic approach, we estimated a longer mean generation time (5.9 days, 5.2–7.0 days) and a similar standard deviation (4.8 days, 4.0–6.3 days). As well as estimating the generation time using data from the entire study period, we also considered whether the generation time varied temporally. Both models suggest a shorter mean generation time in September-November 2020 compared to earlier months. Since the SARS-CoV-2 generation time appears to be changing, further data collection and analysis is necessary to continue to monitor ongoing transmission and inform future public health policy decisions.

**\*For correspondence:**
william.hart@keble.ox.ac.uk

†These authors contributed equally to this work

## Editor's evaluation

This paper is a timely update to the authors previous work and will be of interest to those working on public health responses and the mathematical modelling of infectious diseases. In this work the authors infer the generation interval of SARS–CoV–2 which can allow for the assessment of public health measures. The derivation of the likelihood function is also of interest to mathematical modellers as it allows for the inference of the generation interval from data sets where susceptible depletion may dominate infection dynamics.

## Introduction

The generation time (or generation interval) of a SARS-CoV-2 infector-infectee pair is defined as the period of time between the infector and infectee each becoming infected (*Anderson and May, 1992*; *Diekmann and Heesterbeek, 2000*; *Griffin et al., 2020*; *Svensson, 2007*; *Wallinga and Lipsitch, 2007*). The generation time distribution of many infector-infectee pairs characterises the temporal profile of the transmission risk of an infected host (averaged over all hosts and normalised so that it represents a valid probability distribution; *Fraser, 2007*). Inferring the generation time distribution of SARS-CoV-2 is important in order to predict the effects of non-pharmaceutical interventions such as contact tracing and quarantine (*Ashcroft et al., 2021*; *Ferretti et al., 2020b*; *Hart et al., 2021*). In addition, the generation time distribution is widely used in epidemiological models for estimating the time-dependent reproduction number from case notification data (*Abbott et al., 2020*; *Fraser, 2007*; *Gostic et al., 2020*; *Thompson et al., 2020*) and is crucial for understanding the relationship between the reproduction number and the epidemic growth rate (*Fraser, 2007*; *Parag et al., 2021*; *Park et al., 2020a*; *Wallinga and Lipsitch, 2007*).

The SARS-CoV-2 generation time distribution has previously been estimated using data from known infector-infectee transmission pairs (*Ferretti et al., 2020a*; *Ferretti et al., 2020b*; *Hart et al., 2021*) or entire clusters of cases (*Ganyani et al., 2020*; *Hu et al., 2021*; *Sun et al., 2021*). These studies involved data (*Cheng et al., 2020*; *Ferretti et al., 2020b*; *Ganyani et al., 2020*; *He et al., 2020*; *Xia et al., 2020*; *Zhang et al., 2020*) collected between December 2019 and April 2020, almost entirely from countries in East and Southeast Asia (with the exception of four transmission pairs from Germany and four from Italy in *Ferretti et al., 2020b*). Evidence from January and February 2020 in China suggested a temporal reduction in the mean generation time due to non-pharmaceutical interventions (*Sun et al., 2021*). Specifically, effective isolation of infected individuals is likely to have reduced the proportion of transmissions occurring when potential infectors were in the later stages of infection, thereby shortening the generation time (*Sun et al., 2021*). Similarly, two other studies found a decrease in the serial interval (the difference between symptom onset times of an infector and infectee; *Ali et al., 2020*) and an increase in the proportion of presymptomatic transmissions (*Bushman et al., 2021*) in China over the same time period, which can be attributed to symptomatic hosts being isolated increasingly quickly over time.

Despite estimation of the SARS-CoV-2 generation time in Asia early in the pandemic, relatively little is known about the generation time distribution outside Asia, and whether or not any changes have occurred in the generation time since the early months of the pandemic. At the time of writing, we are aware of only one previous study in which the generation time was estimated using data from the UK (*Challen et al., 2021*). In that study (*Challen et al., 2021*), data describing symptom onset dates for 50 infector-infectee pairs, collected by Public Health England (PHE; now the UK Health Security Agency) between January and March 2020 as part of the 'First Few Hundred' case protocol (*Boddington*

**Table 1.** Previous SARS-CoV-2 generation time estimates.
Estimates of the mean and standard deviation of the generation time distribution, obtained under the assumption of independent transmission and symptoms. 95% credible intervals are shown in brackets where available.

| Study | Location | Time period | Mean generation time (days) | Standard deviation of generation time distribution (days) |
|---|---|---|---|---|
| *Ferretti et al., 2020b* | Various | December 2019-February 2020 | 5.0 | 1.9 |
| *Ganyani et al., 2020* | Singapore | January-February 2020 | 5.20 (3.78–6.78) | 1.72 (0.91–3.93) |
| *Ganyani et al., 2020* | China | January-February 2020 | 3.95 (3.01–4.91) | 1.51 (0.74–2.97) |
| *Hart et al., 2021* | Various | December 2019-March 2020 | 5.57 (5.08–6.09) | 2.32 (1.83–2.91) |
| *Ferretti et al., 2020a* | Various | December 2019-March 2020 | 5.5 | 1.8 |
| *Challen et al., 2021* | UK | January-March 2020 | 4.8 (4.3–5.41) | 1.7 (1.0–2.6) |

*et al., 2021*; *Public Health England, 2020*), were used to infer the generation time distribution. However, since these transmission pairs mostly consisted of international travellers and their household contacts, the authors concluded that their estimates of the generation time may have been biased downwards due to enhanced surveillance and isolation of these cases (*Challen et al., 2021*).

Here, we use data from a household study (*Miller et al., 2021*), conducted between March and November 2020, to estimate the SARS-CoV-2 generation time distribution in the UK under two different underlying transmission models. In the first model (the 'independent transmission and symptoms model'), a parsimonious assumption is made that the generation time and the incubation period of the infector are independent (i.e. there is no link between the times at which infectors transmit the virus and the times at which they develop symptoms), as has often been employed in studies in which the SARS-CoV-2 generation time has been estimated (*Challen et al., 2021*; *Ferretti et al., 2020a*; *Ganyani et al., 2020*; *Hart et al., 2021*; *Lehtinen et al., 2021*; *Table 1*). In the second model (the 'mechanistic model'), we use a mechanistic approach in which potential infectors progress through different stages of infection, first becoming infectious before developing symptoms (*Hart et al., 2021*). Infectiousness is therefore explicitly linked to symptoms in the mechanistic model. A feature of the mechanistic model is that individuals with longer incubation periods will (on average) be infectious for longer before developing symptoms, and so generate more transmissions, compared to those with shorter incubation periods.

By fitting separately to data from three different time intervals within the study period, we explore whether or not there was a detectable temporal change in the generation time distribution.

## Results

### Inferring the generation time from UK household data

We fitted two models of infectiousness (the independent transmission and symptoms model and the mechanistic model) to data collected from 172 UK households in a study (*Miller et al., 2021*) conducted by PHE between March and November 2020 (*Figure 1—source data 1*). Each household was recruited to the study following a confirmed SARS-CoV-2 infection, and all household members were then followed to investigate whether or not they became infected (this was determined through PCR and antibody testing). If a household member was infected and developed symptoms, their symptom onset date was recorded (see Methods).

In our previous work (*Hart et al., 2021*), we fitted the same two models of infectiousness to data from infector-infectee transmission pairs collected in the early months of the COVID-19 pandemic. Here, we adapted the approach presented in that article (*Hart et al., 2021*) in order to estimate the generation time using household transmission data. Specifically, we used data augmentation MCMC, augmenting the observed data with both estimated times of infection and estimated precise times at which symptomatic infected hosts developed symptoms (within recorded symptom onset dates). This enabled us (in the likelihood function) to account for uncertainty about exactly who-infected-whom within a household by summing together likelihood contributions corresponding to infection by different possible infectors. In addition, we corrected for the regularity of household contacts to derive more widely applicable estimates of the generation time. We did this by including a factor in the likelihood to account for each infected individual avoiding infection from household contacts that occurred prior to their actual time of infection (see Methods for full details of our approach).

For the two fitted models, we calculated posterior estimates of the mean (*Figure 1A*) and standard deviation (*Figure 1B*) of the generation time distribution, in addition to the proportion of transmissions occurring prior to symptom onset (among infectors who develop symptoms; *Figure 1C*) and the overall infectiousness parameter, $\beta_0$ (see Methods; *Figure 1D*). Under the commonly used independent transmission and symptoms model, we obtained a point estimate of 4.2 days (95% credible interval (CrI) 3.3–5.3 days) for the mean generation time (*Figure 1A*, blue violin; we calculated point estimates for each model using the posterior means of fitted model parameters because the mode of the joint posterior distribution could not easily be calculated from the output of the MCMC procedure). This value is similar to a previous estimate obtained using data from China by *Ganyani et al., 2020*. It is slightly lower than estimates for Singapore obtained by *Ganyani et al., 2020* and for several countries (predominantly in Asia) obtained by *Ferretti et al., 2020b* (*Table 1*), although those estimates lie within our credible interval. On the other hand, our estimated standard deviation

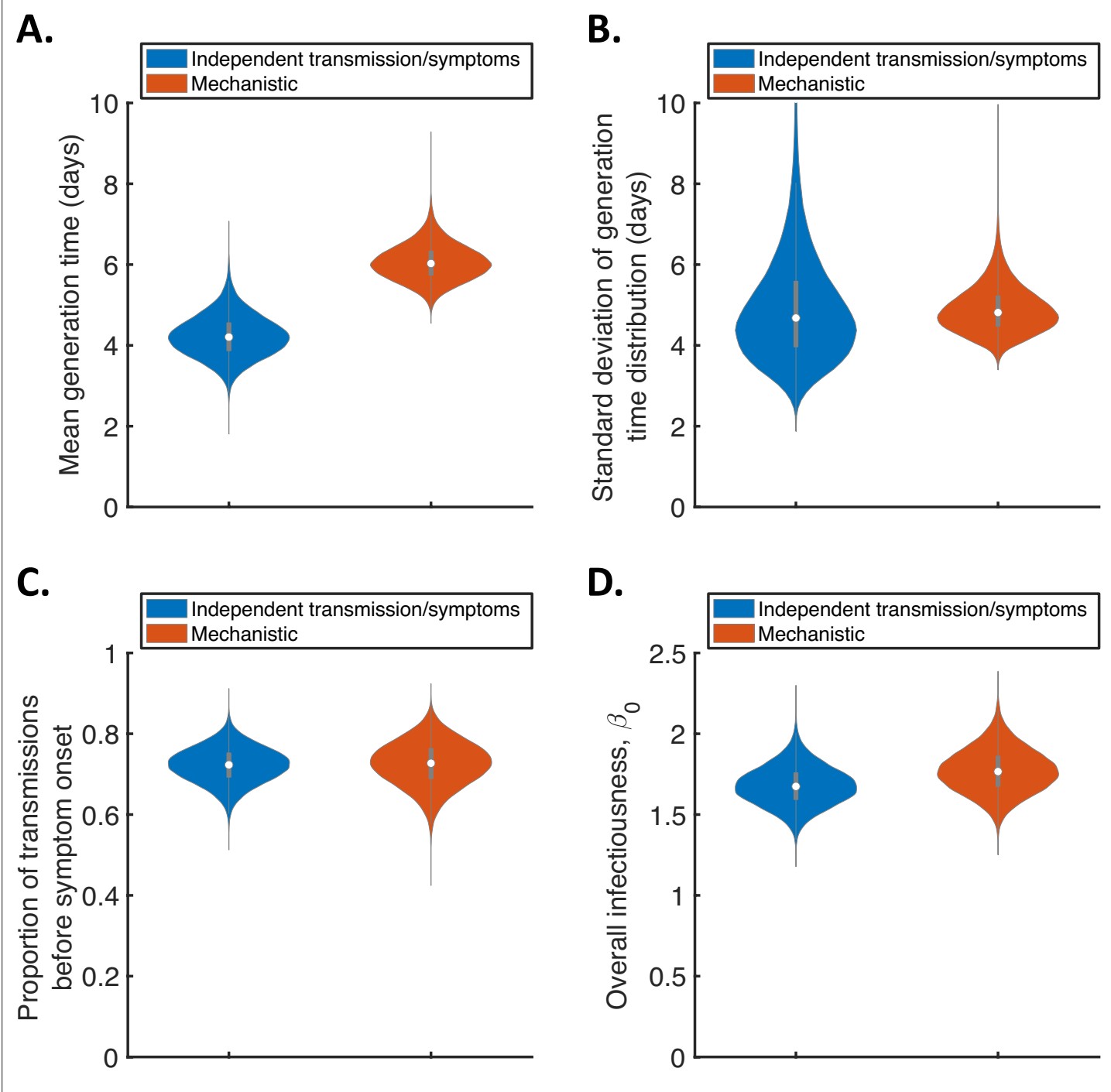

**Figure 1.** Comparison of posterior predictions. Violin plots indicating posterior distributions of the mean (**A**) and standard deviation (**B**) of the generation time distribution, proportion of transmissions occurring prior to symptom onset (among infectors who develop symptoms; **C**), and overall infectiousness parameter, $\beta_0$ (describing the expected number of household transmissions generated by a single infected host) in a large, otherwise entirely susceptible, household; **D**). We show results obtained both using a model in which infectiousness is assumed to be independent of when symptoms develop ('independent transmission and symptoms model', blue), and using the mechanistic model from *Hart et al., 2021* in which infectiousness is explicitly linked to symptoms ('mechanistic model', red).

The online version of this article includes the following source data and figure supplement(s) for figure 1:

**Source data 1.** Household transmission data.

**Figure supplement 1.** Posterior distributions of fitted parameters for the independent transmission and symptoms model.

*Figure 1 continued on next page*

of 4.9 days (95% CrI 3.0–8.3 days; *Figure 1B*, blue violin) is substantially higher than previous estimates (*Table 1*). Using our mechanistic model, we obtained a higher estimate for the mean generation time of 5.9 days (95% CrI 5.2–7.0 days; *Figure 1A*, red violin), and a similar estimate for the standard deviation (4.8 days, 95% CrI 4.0–6.3 days; *Figure 1B*, red violin), compared to those predicted by the independent transmission and symptoms model.

The two models gave similar posterior distributions for the proportion of transmissions prior to symptom onset (*Figure 1C*). Specifically, point estimate values of model parameters led to an estimated proportion of transmissions prior to symptom onset of 0.72 (95% CrI 0.63–0.80) for the independent transmission and symptoms model, and 0.73 (95% CrI 0.61–0.83) for the mechanistic model. These estimates are higher than obtained in some previous studies in which the infectiousness profile of SARS-CoV-2 infected hosts at each time since infection and/or time since symptom onset has been estimated (*Ashcroft et al., 2020*; *Ferretti et al., 2020a*; *He et al., 2020*). On the other hand, our point estimates for the two models both lie within the 95% credible interval obtained for the mechanistic model in our previous work (0.53–0.77, point estimate 0.65; *Hart et al., 2021*). Similar or higher estimates also exist in the wider literature (*Casey-Bryars et al., 2021*; *Ganyani et al., 2020*; *Tindale et al., 2020*).

Posterior distributions for fitted model parameters are shown in *Figure 1—figure supplement 1* and *Figure 1—figure supplement 2*, and point estimates and 95% credible intervals are given in *Appendix 1—table 2* and *Appendix 1—table 3*. Since only the likelihood with respect to augmented data was calculated in the MCMC procedure, direct comparisons of the goodness of fit between the models were not readily available. However, comparing model predictions of the distribution of the interval between successive symptom onset dates in households to the analogous distribution in the data indicated that both models provided a similar fit to the data (*Figure 1—figure supplement 3*).

In *Figure 1* (and elsewhere, unless otherwise stated), we characterise the generation time distribution assuming that a constant supply of susceptible individuals are available to infect during the course of infection. This distribution corresponds to the normalised expected infectiousness profile of an infected host at each time since infection, and is widely applicable to transmission outside of, as well as within, households. However, realised household generation times are expected to be shorter than the estimates shown in *Figure 1*. This is due to the depletion of susceptible household members before longer generation times can be obtained, especially in small households (*Cauchemez et al., 2009*; *Fraser, 2007*; *Park et al., 2020a*). As a result, we also predicted the mean and standard deviation of realised generation times within the study households (*Figure 1—figure supplement 4A,B*), accounting for the precise distribution of household sizes in the study. For both the independent transmission and symptoms model and the mechanistic model, the mean (point estimates 3.6 days and 4.9 days for the two models, respectively) and standard deviation (3.8 days and 4.1 days) of realised household generation times were lower than our main generation time estimates shown in *Figure 1*. Since household transmission typically occurs earlier in the infector's course of infection than indicated by the estimates shown in *Figure 1*, we predicted a higher proportion of presymptomatic

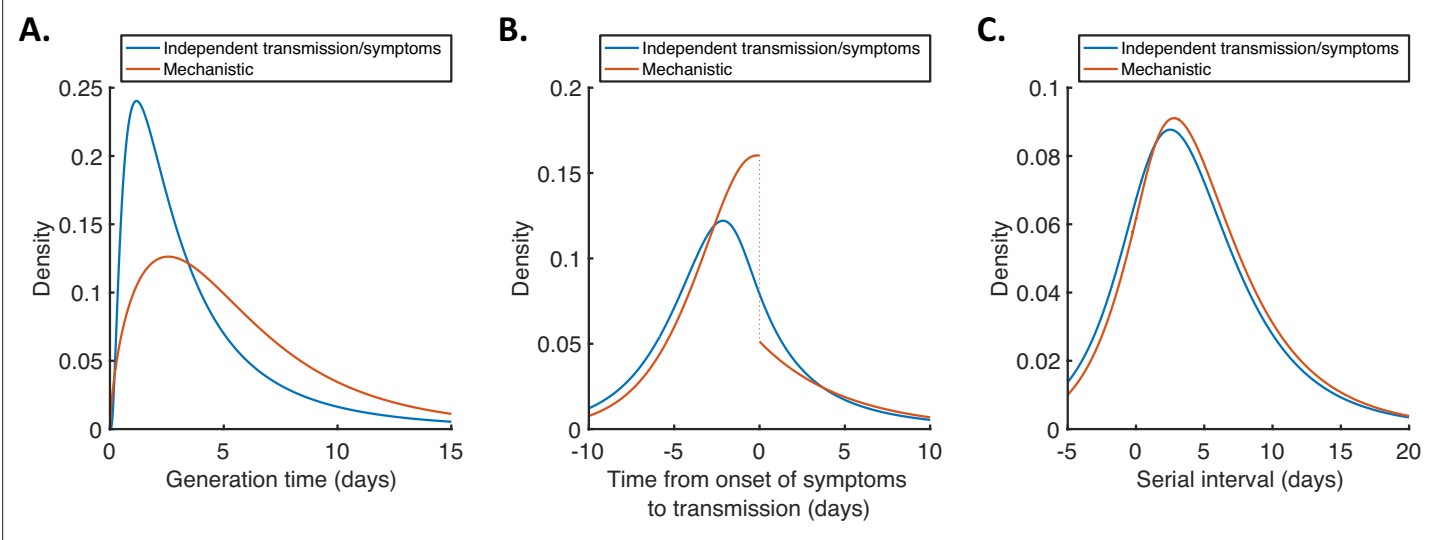

**Figure 2.** Generation time, TOST and serial interval distributions. Inferred generation time (**A**), TOST (**B**) and serial interval (**C**) distributions for the two models, obtained using point estimate (posterior mean) parameters. The means and standard deviations of these distributions are given in *Appendix 1—table 4*. Similarly to *Hart et al., 2021*, the discontinuity in the red curve in (**B**) occurs because different transmission rates were fitted for infectors in the presymptomatic infectious (*P*) and symptomatic infectious (*I*) stages of infection. The reduction in transmission following symptom onset can be attributed to changes in behaviour in response to symptoms (*Manfredi and D'Onofrio, 2013*).

transmissions within the study households (*Figure 1—figure supplement 4C*) compared to the estimates in *Figure 1C*.

For both models, we then used point estimates of fitted model parameters to infer the distributions of the generation time (*Figure 2A*), the time from onset of symptoms to transmission (TOST; *Figure 2B*) and the serial interval (*Figure 2C*). The TOST distribution (which characterises the relative expected infectiousness of a host (who develops symptoms) at each time from symptom onset, as opposed to from infection [*Ashcroft et al., 2020*; *Ferretti et al., 2020a*; *He et al., 2020*; *Lehtinen et al., 2021*; *Wells et al., 2021*]) obtained using the mechanistic model was more concentrated around the time of symptom onset compared to that predicted assuming independent transmission and symptoms (*Figure 2B*), as we found in our previous work (*Hart et al., 2021*). In contrast, the estimated serial interval distributions were similar for the two models (*Figure 2C*). The means and standard deviations of the distributions shown in *Figure 2* are given in *Appendix 1—table 4*.

## Temporal variation in the generation time distribution

To explore whether or not the generation time distribution changed during the study period, we separately fitted the independent transmission and symptoms model to the data from households in which the index case was recruited in (i) March-April, (ii) May-August, or (iii) September-November 2020 (*Figure 3*). We chose these time periods to ensure the numbers of households recruited into the study during each interval were similar (*Figure 3—figure supplement 1*).

The results shown in *Figure 3A* suggest a shorter mean generation time in September-November 2020 (2.9 days, 95% CrI 1.8–4.3 days) compared to earlier months (4.9 days, 95% CrI 3.6–6.3 days, for March-April and 5.2 days, 95% CrI 3.4–7.2 days, for May-August). Comparing the posterior estimates for May-August and September-November (the red and orange violins in *Figure 3A*, respectively) indicated a 97% posterior probability of a shorter mean generation time in the later of these two time periods. A similar temporal reduction in the mean generation time was found when we instead fitted the mechanistic model to the data from the three time intervals (*Figure 3—figure supplement 2*). Estimates of the mean generation time using the mechanistic model were 6.5 days (95% CrI 5.6–8.1 days) for March-April, 7.1 days (95% CrI 5.7–9.6 days) for May-August, and 5.1 days (95% CrI 4.3–6.4 days) for September-November, with a 98% posterior probability of a shorter mean generation time in September-November than May-August. We also used point estimates of model parameters to compare the distributions of the generation time, TOST and serial interval between

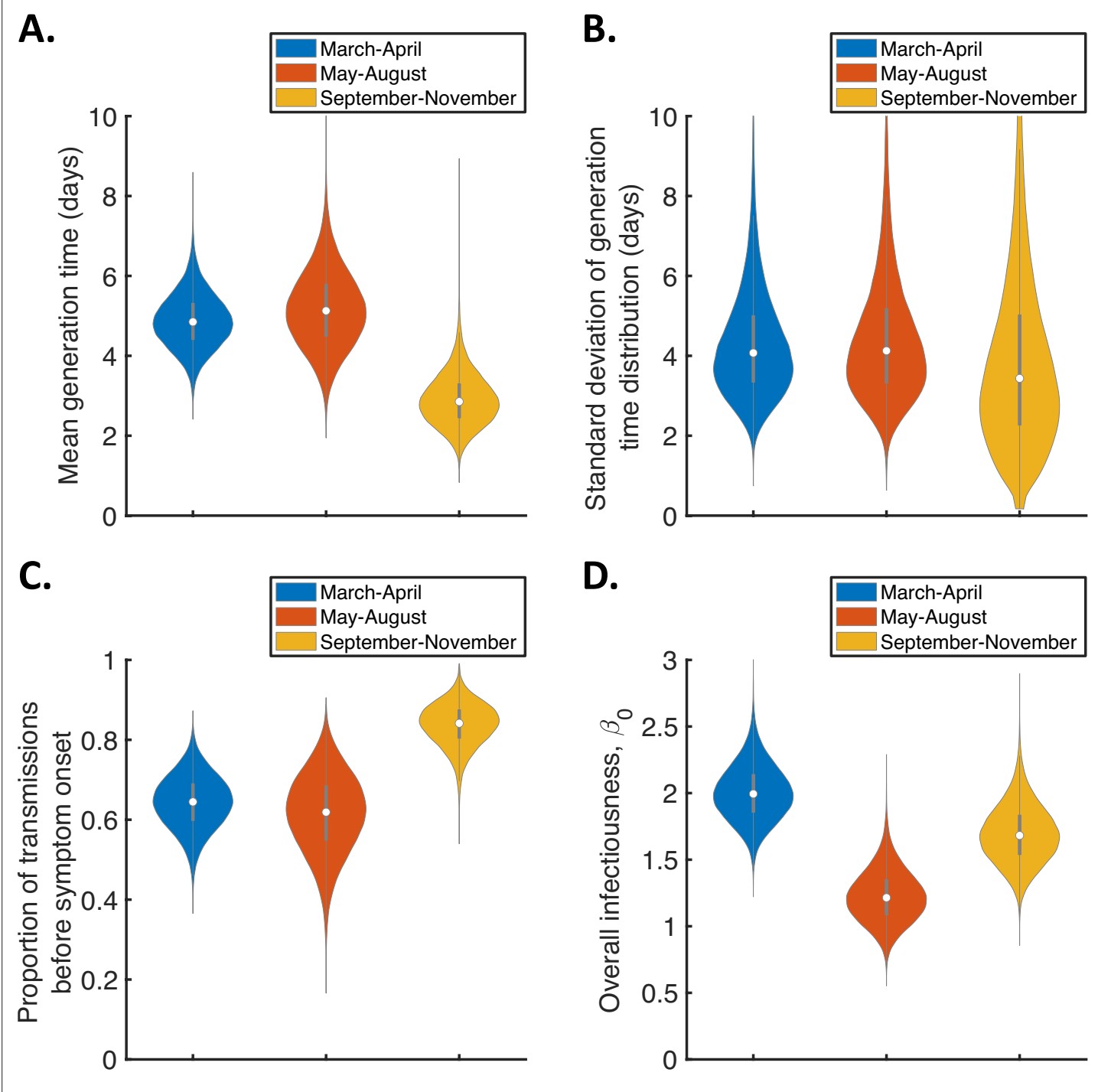

**Figure 3.** Temporal changes in the generation time. Violin plots indicating posterior distributions of the mean (**A**) and standard deviation (**B**) of the generation time distribution, proportion of transmissions occurring prior to symptom onset (among infectors who develop symptoms; **C**), and overall infectiousness parameter, $\beta_0$ (**D**), for the independent transmission and symptoms model fitted to data from March-April (blue), May-August (red), or September-November 2020 (orange).

The online version of this article includes the following figure supplement(s) for figure 3:

**Figure supplement 1.** Number of study households and household members by recruitment month.

**Figure supplement 2.** Temporal changes in the generation time for the mechanistic model.

**Figure supplement 3.** Temporal changes in the generation time, TOST and serial interval distributions.

*Figure 3 continued on next page*

*Figure 3 continued*

**Figure supplement 4.** Monthly changes in the generation time from September-November 2020 for the independent transmission and symptoms model.

**Figure supplement 5.** Temporal changes in fitted parameters for the mechanistic model.

**Figure supplement 6.** Temporal changes in the generation time for the independent transmission and symptoms model, accounting for the possibility of co-primary cases.

the time periods (*Figure 3—figure supplement 3*), with both models indicating that the transmission risk peaked earlier in infection for individuals infected in September-November compared to earlier months (*Figure 3—figure supplement 3A,D*).

*Figure 3C* shows posterior estimates for the proportion of transmissions occurring prior to symptom onset (among symptomatic infectors) across the three time periods for the independent transmission and symptoms model, indicating a very high proportion of presymptomatic transmissions in September-November (0.83, 95% CrI 0.72–0.93) compared to lower estimates for March-April (0.64, 95% CrI 0.51–0.77) and May-August (0.62, 95% CrI 0.41–0.79). Our results for the mechanistic model indicate a similar temporal increase in the proportion of presymptomatic transmissions during the study period (*Figure 3—figure supplement 2C*).

To explore the lower estimated generation time for September-November further, we also fitted the independent transmission and symptoms model to the data from each of these months individually (*Figure 3—figure supplement 4*). The shorter estimated generation time compared to earlier in the pandemic was consistent across each of the three months (*Figure 3—figure supplement 4A*). We note that, while the Alpha (B.1.1.7) variant had begun to emerge in the UK by the end of the study period (*Public Health England, 2021*), genomic surveillance as part of the study showed that this variant caused infections in only two study households. This variant was therefore unlikely to have been responsible for the temporal reduction in the generation time that we observed.

In *Figure 3—figure supplement 5*, we show the posterior distributions of the fitted parameters for the mechanistic model (other than the overall infectiousness, $\beta_0$, which is shown in *Figure 3D*) over the different time periods. These parameters represent the mean duration of the platent period (expressed as a proportion of the mean incubation period; *Figure 3—figure supplement 5A*), the mean duration of the symptomatic infectious period (*Figure 3—figure supplement 5B*), and the relative infectiousness of presymptomatic infectious hosts compared to those with symptoms (*Figure 3—figure supplement 5C*). However, there was substantial overlap in the credible intervals of posterior estimates of each parameter between the three time periods. We were therefore unable to identify the precise parameter(s) responsible for the decrease in generation time and increase in the proportion of presymptomatic transmissions that we observed.

## Sensitivity analyses

When we fitted the two models to the household transmission data, we assumed that each household transmission chain was initiated by a single primary case and all other infected household members were infected from within the household. However, we also extended our framework to account for the possibility of co-primary cases (Appendix 1, *Figure 1—figure supplement 5* and *Figure 3—figure supplement 6*). This led to slightly higher estimates of the mean generation time (*Figure 1—figure supplement 5A*) under each model compared to the corresponding estimates shown in *Figure 1A*, with point estimates of 4.8 days (95% CrI 3.6–6.3 days) for the independent transmission and symptoms model and 6.8 days (95% CrI 5.7–8.6 days) for the mechanistic model. Estimates of the standard deviation of the generation time distribution were similar to those in *Figure 1* (*Figure 1—figure supplement 5B*); point estimates were 4.8 days (95% CrI 2.9–7.9 days) for the independent transmission and symptoms model and 5.1 days (95% CrI 4.0–6.9 days for the mechanistic model). As part of the fitting procedure, we estimated the probability that each household member was infected during the primary transmission event (*Figure 1—figure supplement 5E*), obtaining point estimates of 0.17 (95% CrI 0.02–0.33) under the independent transmission and symptoms model and 0.27 (95% CrI 0.10–0.41) under the mechanistic model. We also repeated the analyses in *Figure 3* but accounting for the possibility of co-primary cases (*Figure 3—figure supplement 6*). Our main qualitative finding remained unchanged: the mean generation time was found to decrease during the study period (*Figure 3—figure supplement 6A*).

In the independent transmission and symptoms model, we assumed that both the generation time and incubation period follow lognormal distributions. The mean and standard deviation of the generation time distribution were estimated by fitting the model to the household transmission data. In the fitting procedure, we assumed that the incubation period followed a lognormal distribution that was obtained in a previous meta-analysis (*McAloon et al., 2020*). In contrast, we assumed in our mechanistic approach that each infection could be decomposed into three gamma distributed stages (latent, presymptomatic infectious and symptomatic infectious), so that the incubation period was also gamma distributed (with the same mean and standard deviation as the lognormal distribution obtained by *McAloon et al., 2020*). An expression for the generation time distribution in the mechanistic model, which does not take a simple parametric form, is given in the Appendix. However, we conducted supplementary analyses in which we instead assumed that either the generation time (*Figure 1—figure supplement 6*) or incubation period (*Figure 1—figure supplement 7*) in the independent transmission and symptoms model was gamma distributed. In both cases, we obtained similar results to those shown for that model in *Figure 1*.

We also relaxed the assumption of a fixed incubation period distribution (*Figure 1—figure supplement 8*), using the confidence intervals obtained by *McAloon et al., 2020* to account for uncertainty in the incubation period distribution (*Figure 1—figure supplement 8A, B*). For both the independent transmission and symptoms model and the mechanistic model, accounting for this uncertainty did not substantially affect posterior estimates of either the mean (*Figure 1—figure supplement 8C*) or the standard deviation (*Figure 1—figure supplement 8D*) of the generation time distribution.

In our main analyses, we assumed that household transmission was frequency-dependent, so that the force of infection exerted by an infected household member on each susceptible household member scales with $1/n$, where $n$ is the household size (*Cauchemez et al., 2014*; *Cauchemez et al., 2004*). However, since some studies of influenza virus transmission in households have found transmission to lie somewhere in between frequency- and density-dependent (*Endo et al., 2019*; *Ferguson et al., 2005*), we also considered alternative possibilities where infectiousness scales with $n^{-\rho}$, for different values of $\rho$. In *Figure 1—figure supplement 9A-C*, we compared estimates under our baseline value of $\rho = 1$ (frequency-dependent transmission) with those obtained assuming either $\rho = 0$ (density-dependent transmission) or the intermediate possibility of $\rho = 0.5$ considered by *Endo et al., 2019*. In addition, we conducted an analysis in which the dependency, $\rho$, was estimated alongside other model parameters (*Figure 1—figure supplement 9D*). We found that our estimates of the mean and standard deviation of the generation time distribution were robust to the assumed value of $\rho$ (*Figure 1—figure supplement 9A, B*). However, when the value $\rho$ was fitted (*Figure 1—figure supplement 9D*), we estimated a value of 1.0 (95% CrI 0.6–1.5). This supported our assumption of frequency-dependent transmission, although the credible interval was relatively wide. In addition, we considered the possibility that infectiousness instead scales with $1/(n - 1)$, so that the infector under consideration is not included in this scaling, and again obtained similar estimates of the mean and standard deviation of the generation time distribution compared to those shown in *Figure 1* (*Figure 1—figure supplement 10*).

We also considered the sensitivity of our results to the assumed relative infectiousness of asymptomatic infected hosts (*Figure 1—figure supplement 11*). In most of our analyses, we assumed that the expected infectiousness of an infected host who remained asymptomatic throughout infection was a factor $\alpha_A = 0.35$ times that of a host who develops symptoms, at each time since infection (*Buitrago-Garcia et al., 2020*). However, similar estimates of the mean (*Figure 1—figure supplement 11*) and standard deviation (*Figure 1—figure supplement 11B*) of the generation time distribution were obtained when we instead assumed $\alpha_A = 0.1$ or $\alpha_A = 1.27$ (these values corresponded to the lower and upper confidence bounds obtained by *Buitrago-Garcia et al., 2020*). Lower values of $\alpha_A$ did lead to slightly higher estimates of the overall infectiousness of infectors who develop symptoms, $\beta_0$ (*Figure 1—figure supplement 11D*). However, this effect was minimal, likely because very few cases in the household study were asymptomatic (27 out of 357).

Finally, we explored the robustness of our results to the exclusion of household members of unknown infection status (see Methods), considering the extreme possibilities where these individuals were instead assumed to have either all remained uninfected, or all become infected (*Figure 1—figure supplement 12*). Although the estimates of $\beta_0$ were affected by this assumption (*Figure 1—figure*

*supplement 12D*), the estimated generation time distribution was robust to the assumed infection status of these individuals (*Figure 1—figure supplement 12A,B*).

## Discussion

In this study, we estimated the generation time distribution of SARS-CoV-2 in the UK by fitting two different models to data describing the infection status and symptom onset dates of individuals in 172 households. The first model was predicated on an assumption that transmission and symptoms are independent. While this assumption has often been made in previous studies in which the SARS-CoV-2 generation time has been estimated (*Challen et al., 2021*; *Deng et al., 2021*; *Ferretti et al., 2020b*; *Ganyani et al., 2020*; *Knight and Mishra, 2020*), it is not an accurate reflection of the underlying epidemiology (*Bacallado et al., 2020*; *Lehtinen et al., 2021*). Therefore, we also considered a mechanistic model based on compartmental modelling, which was shown in our earlier work (*Hart et al., 2021*) to provide an improved fit to data from 191 SARS-CoV-2 infector-infectee pairs compared to previous models that have been used to estimate the generation time. Here, infection times and the order of transmissions within households were unknown, whereas in *Hart et al., 2021* the direction of transmission was assumed to be known for each infector-infectee pair. For that reason, we needed to extend the statistical inference methods underlying our previous work (*Hart et al., 2021*) to fit the two models to household data. To do this, we used a data augmentation MCMC approach similar to previous studies of household influenza virus transmission (*Cauchemez et al., 2009*; *Cauchemez et al., 2004*; *Ferguson et al., 2005*).

Under the model assuming independent transmission and symptoms, we estimated a mean generation time of 4.2 days (95% CrI 3.3–5.3 days) and a standard deviation of 4.9 days (95% CrI 3.0–8.3 days). The estimate of the mean generation time was comparable to previous estimates obtained under this assumption using data from elsewhere (*Ferretti et al., 2020a*; *Ferretti et al., 2020b*; *Ganyani et al., 2020*; *Table 1*). On the other hand, while our credible interval for the standard deviation was wide, the estimates obtained in those previous studies (*Ferretti et al., 2020a*; *Ferretti et al., 2020b*; *Ganyani et al., 2020*) all lay below our lower 95% credible limit of 3.0 days. One potential cause of this disparity is the difference in isolation policies for symptomatic hosts between countries. In particular, the UK's policy of self-isolation may be expected to lead to a longer-tailed generation time distribution compared to countries with a policy of isolation outside the home, since under home isolation, some within-household transmission is likely to occur even following isolation. Isolation outside the home was commonplace in the East and Southeast Asian countries where the majority of the data underlying the estimates by *Ferguson et al., 2005*; *Ferretti et al., 2020a*; *Ganyani et al., 2020* were collected.

Using the mechanistic model, we predicted a higher mean generation time of 5.9 days (95% CrI 5.2–7.0 days) compared to the value estimated under the assumption of independent transmission and symptoms. On the other hand, the inferred serial intervals for the independent transmission and symptoms model and mechanistic model were more similar (*Figure 2C*), with means of 4.2 days and 4.7 days, respectively. Temporal information in our household transmission data consisted mostly of symptom onset dates, with very few individuals testing positive before developing symptoms. Therefore, the variation in estimates of the generation time between the models can be attributed to differences in the assumed relationships between the generation time and serial interval under those models. For the independent transmission and symptoms model, the generation time and serial interval distributions have the same mean, as is commonly assumed to be the case (*Lehtinen et al., 2021*). However, this was not true for the mechanistic model, in which infected hosts with longer presymptomatic infectious periods generate (on average) a higher number of transmissions. As a result, under the mechanistic model, a randomly chosen infection is more likely to arise from an infector with a longer incubation period than from a host with a shorter incubation period, thereby leading to a longer generation time than serial interval (an analytical expression for the exact difference between the mean generation time and serial interval for that model is derived in the Appendix).

Our results do not indicate any clear difference in goodness of fit to the data between the two models (*Figure 1—figure supplement 3*). A range of factors should therefore be considered when deciding which of our estimates of epidemiological parameters to use in subsequent analyses. Although any model requires simplifying assumptions to be made, our mechanistic approach allows the standard assumption of independent transmission and symptoms to be relaxed by providing a

mechanistic underpinning to the relationship between the times at which individuals display symptoms and become infectious. Furthermore, as described above, this model was shown in our previous work (*Hart et al., 2021*) to provide a better fit to an earlier SARS-CoV-2 dataset than a model assuming independence between transmission and symptoms (in our earlier work [*Hart et al., 2021*], the simpler setting of transmission pairs rather than households facilitated direct model comparison). On the other hand, the independent transmission and symptoms model has the advantage of producing an estimated generation time distribution with a simple parametric form. The choice of estimates to use may also depend on precisely what the estimates are being used for. For example, the generation time distribution inferred under the assumption of independent transmission and symptoms may be better suited for use in some models for estimating the time-dependent reproduction number, since those models often also involve the assumption that transmission and symptoms are independent (*Abbott et al., 2020*). In contrast, the parameter estimates from our mechanistic approach correspond naturally to parameters in compartmental epidemic models.

By fitting separately to data from three different intervals within the study period (March-November 2020), we investigated whether or not the generation time distribution in the UK changed as the pandemic progressed. Our results indicate a shorter mean generation time in September-November compared to earlier months (*Figure 3A*). One possible explanation for this is a higher proportion of time spent indoors in colder months leading to an increased transmission risk, particularly in the early stages of infection before symptoms develop (since symptomatic infected hosts are still likely to self-isolate). This explanation is consistent with our finding in *Figure 3C* of a higher proportion of transmissions occurring prior to symptom onset in September-November compared to March-April and May-August.

While behavioural changes may have been responsible for our finding of a temporal decrease in the generation time, an alternative explanation could be that evolutionary changes in the SARS-CoV-2 virus that occurred during the study period affected the generation time. For example, the B.1.177 lineage emerged in Spain in early summer 2020, and became the dominant SARS-CoV lineage in the UK around the beginning of October 2020 (*Vöhringer et al., 2021*). Subsequently, the Alpha (B.1.1.7) variant, which was first detected in September 2020, became dominant in the UK in December 2020 (*Public Health England, 2021*). The Alpha variant has been shown to possess different characteristics than earlier variants (*Davies et al., 2021*; *Volz et al., 2021*), causing an increased epidemic growth rate in the UK that has been attributed to an increase in transmissibility of 43%–90% (*Davies et al., 2021*). While in principle evolutionary changes could explain the variation in the generation time that we observed, sequencing data show that the Alpha variant was responsible for infections in only two households within our dataset. Consequently, the Alpha variant was not responsible for our main finding of a temporally decreasing generation time, and additional data are required to quantify the impact of the emergence of that variant (and subsequent variants, such as the Delta (B.1.617.2) and Omicron (B.1.1.529) variants) on the SARS-CoV-2 generation time.

In data collected from infector-infectee transmission pairs, shorter generation times are expected to be over-represented at times when case numbers are rising (*Britton and Scalia Tomba, 2019*; *Ferretti et al., 2020b*; *Lehtinen et al., 2021*), and vice versa. While we used data from households (rather than transmission pairs) in our analyses, a similar effect may have contributed to our shorter estimated mean generation time for September-November 2020 (national case numbers were mostly increasing in September-October 2020) compared to earlier months of the study (during which case numbers were mostly decreasing; *Knock et al., 2021*; *Pouwels et al., 2021*). However, we estimated the mean generation time to be similar in November (when case numbers were mostly decreasing [*Knock et al., 2021*; *Pouwels et al., 2021*]) compared to September and October (*Figure 3—figure supplement 4*), suggesting that this effect of background epidemic dynamics alone did not drive the temporal changes in generation time that we observed. We note, however, that sample sizes for individual months were small (*Figure 3—figure supplement 1*). Extending our household inference framework to explicitly account for background epidemic dynamics in generation time estimates (similar to methods that have been developed for transmission pair data [*Britton and Scalia Tomba, 2019*; *Ferretti et al., 2020b*]) is an avenue for future work.

Our finding of a temporal decrease in the mean generation time during the study period highlights the importance of obtaining up-to-date generation time estimates specific to the location under study. Should variations in the generation time distribution occur and not be accounted for, estimates of the

time-dependent reproduction number may be incorrect (*Park et al., 2021*; *Wallinga and Lipsitch, 2007*). Specifically, if the mean generation time is shorter than assumed, then the true value of the time-dependent reproduction number is likely to be closer to one than the inferred value (*Wallinga and Lipsitch, 2007*), and vice versa.

One advantage of our approach compared to previous studies in which the SARS-CoV-2 generation time has been estimated (*Ferretti et al., 2020a*; *Ganyani et al., 2020*; *Hart et al., 2021*) is that we were able to include the contribution of asymptomatic infected hosts to household transmission chains in our analyses. We showed that our estimated generation time distribution was robust to the assumed relative infectiousness of infected hosts who remain asymptomatic, $\alpha_A$ (*Figure 1—figure supplement 11*). Similarly, while we assumed frequency-dependent household transmission in most of our analyses, we found that the exact relationship between the household size and transmission had little effect on our estimates of the mean and standard deviation of the generation time distribution (*Figure 1—figure supplement 9* and *Figure 1—figure supplement 10*). We also considered estimating the exponent governing the dependency of transmission on household size (*Figure 1—figure supplement 9D*). This supported our assumption of frequency-dependent transmission, and is consistent with the finding of an inverse relationship between household size and secondary attack rate in the household study underlying our analyses (*Miller et al., 2021*). In previous studies of influenza transmission within households, evidence has been found both in favour of (*Cauchemez et al., 2004*) and against (*Endo et al., 2019*) frequency-dependent transmission.

While our generation time estimates were robust to the assumed relative infectiousness of infected hosts who remain asymptomatic and whether transmission was assumed to be frequency- or density-dependent, extending our approach to account for the possibility that household transmission chains originate with multiple co-primary cases led to slightly higher estimates of the generation time (*Figure 1—figure supplement 5*) compared to our main estimates (*Figure 1*). Despite the overall higher estimated generation time, our main qualitative finding of a temporal decrease in the generation time held when co-primary cases were incorporated (*Figure 3—figure supplement 6*).

Like any mathematical modelling study, our approach has some limitations. We used household data in our analyses, whereas some characteristics of wider community transmission may differ from those of transmission within households. However, we corrected for the regularity of household contacts to estimate the (expected) infectiousness profile of an infected host at each time since infection (accounting for behavioural factors), which provides a widely applicable generation time estimate (*Figure 1*). Specifically, the infectiousness profile gives the generation time distribution under the assumption that a constant supply of susceptible individuals are available throughout the course of infection. This distribution can then be conditioned to specific population structures, as we demonstrated by estimating the realised generation time distribution within the study households (*Figure 1—figure supplement 4*). The household generation time estimates shown in *Figure 1—figure supplement 4* are shorter than our main generation time estimates (*Figure 1*), due to the regularity of household contacts and the depletion of susceptible individuals within households before longer generation times can be realised.

We also note that, while our dataset involved a larger sample size than used in most other studies in which the SARS-CoV-2 generation time was estimated (*Ferretti et al., 2020a*; *Ferretti et al., 2020b*; *Ganyani et al., 2020*; *Hart et al., 2021*), the demographics of the study households may not have been completely representative of the wider population. Exploring heterogeneity in the generation time distribution between individuals and/or households with different characteristics is an important topic for future work. This could involve, but is not limited to, estimating the generation time distribution for individuals of different age, sex, ethnicity, and socio-economic status. Nonetheless, as well as providing updated SARS-CoV-2 generation time estimates, our study demonstrates that changes in the generation time can be detected using data from household studies. Our finding that the generation time has become shorter highlights both the importance of continued monitoring of the generation time and the role of household studies in such monitoring efforts, particularly in light of the more recent emergence of novel SARS-CoV-2 variants.

In summary, we have inferred the SARS-CoV-2 generation time distribution in the UK using household data and two different transmission models. A key output of this research is one of the first estimates of the SARS-CoV-2 generation time outside Asia. Another crucial feature of our analysis is that it was based on data from beyond the first few months of the pandemic. Since this research

suggests that the generation time may be changing, continued data collection and analysis is of clear importance.

## Methods

### Data

Data were obtained from a household study (*Miller et al., 2021*) conducted in 172 UK households (with 603 household members in total) by PHE between March and November 2020 (*Figure 1—source data 1*). In each household, an index case was recruited following a positive PCR test. The following were then recorded for each household member:

- The timing and outcome of (up to) two subsequent PCR tests.
- The outcome of an antibody test (carried out for 541 individuals – 90% of the study cohort).
- Whether or not the household member developed symptoms.
- The date of symptom onset (only for symptomatic individuals with a positive PCR or antibody test).

In the study, all household members who tested positive in either a PCR or antibody test were assumed to have been infected. Conversely, all individuals who tested negative for antibodies and did not return a positive PCR test (i.e. the two PCR tests were either negative or were not carried out) were assumed to have remained uninfected, irrespective of symptom status. For 34 individuals (6% of the study cohort), no antibody test was carried out and any PCR tests were negative. Since the available data were considered insufficient to determine whether or not these 34 individuals were infected, these individuals were excluded from our main analyses (but were counted in the household size), although we also considered the sensitivity of our results to this assumption.

In two households, at least one household member developed symptoms 55–56 days prior to the symptom onset date of the index case, with no other household members developing symptoms (or returning a positive PCR or antibody test) between these dates. In contrast, the maximum gap between successive symptom onset dates in the remaining households was 25 days (*Figure 1—figure supplement 3*). Data from these two households were excluded from our analyses, on the basis that the virus was most likely introduced multiple times into these households. Three other households were also excluded from our analyses because, other than the index cases in each household, all other household members were of unknown infection status (i.e. they were among the individuals for whom no antibody test was carried out and any PCR tests were negative).

Overall, aside from the five excluded households, the 167 remaining households comprised 587 individuals, of whom 330 became infected and developed symptoms, 27 became infected but remained asymptomatic, 200 remained uninfected, and the remaining 30 were of unknown infection status. The number of households and individuals recruited into the study by month is shown in *Figure 3—figure supplement 1*.

### Models

#### General modelling framework

Throughout, we denote the expected force of infection exerted by an infected host onto each susceptible member of their household, at time since infection $\tau$, by $\beta(\tau)$, where we assumed

$$\beta(\tau) = \left(\beta_0/n\right)f(\tau),$$

for a host who develops symptoms, and

$$\beta(\tau) = \alpha_A\left(\beta_0/n\right)f(\tau),$$

for a host who remains asymptomatic throughout infection. Here:

- $\beta_0$ is the overall infectiousness parameter, describing the expected number of household transmissions generated by a single infected host (who develops symptoms) in a large, otherwise entirely susceptible, household.
- $n$ is the household size. The scaling of $\beta(\tau)$ with $1/n$ corresponds to frequency-dependent transmission, as assumed by *Cauchemez et al., 2014*; *Cauchemez et al., 2004*, although we carried out a sensitivity analysis in which we considered alternative possibilities where household

transmission is density-dependent (without the scaling factor $1/n$), scales with $1/n^{0.5}$ (***Endo et al., 2019***), or scales with $1/(n-1)$.

- $f(\tau)$ is the generation time distribution (which was assumed to be the same for entirely asymptomatic hosts as those who develop symptoms).
- $\alpha_A$ is the relative infectiousness of infected hosts who remain asymptomatic throughout infection. We assumed a value of 0.35 (***Buitrago-Garcia et al., 2020***) in most of our analyses, although we considered different values of $\alpha_A$ in a sensitivity analysis.

Except where otherwise stated, we considered the generation time distribution assuming a constant supply of susceptibles during infection, $f(\tau)$, which corresponds to the normalised expected infectiousness profile and gives a widely applicable generation time estimate (see Discussion). However, realised generation times within a household may be shorter than predicted by this distribution due to the depletion of susceptible household members before longer generation times can be realised (***Cauchemez et al., 2009***; ***Fraser, 2007***; ***Park et al., 2020b***). For example, if infected hosts are (on average) equally infectious at two times since infection, $\tau_1 < \tau_2$, then $f(\tau_1) = f(\tau_2)$. However, because the number of susceptible household members may decrease between these two times (i.e. either the host under consideration, or another infected household member, may transmit the virus within the household in the intervening time), then transmission is in fact more likely to occur in a household at the earlier time, $\tau_1$, when more susceptibles are available. Therefore, we also predicted the mean and standard deviation of realised generation times within the study households in ***Figure 1—figure supplement 4***.

We considered two different models of infectiousness, which are outlined below. Under each model, expressions were derived in ***Hart et al., 2021*** for the generation time, TOST and serial interval distributions, in addition to the proportion of transmissions occurring before symptom onset. These expressions are given in the Appendix here (other than the generation time distribution and proportion of presymptomatic transmissions for the independent transmission and symptoms model, which are stated below).

## Independent transmission and symptoms model

In this model, the infectiousness of an infected host (who does not remain asymptomatic throughout infection; asymptomatic infected hosts are considered separately) at a given time since infection, $\tau$, is assumed to be independent of exactly when the host develops symptoms – that is, the generation time and incubation period are independent. In our main analyses using this model, we assumed that the generation time distribution, $f(\tau)$, is the probability density function of a lognormal distribution (***Ferguson et al., 2005***; an alternative case of a gamma distributed generation time is considered in ***Figure 1—figure supplement 6***). The mean and standard deviation of this distribution, in addition to $\beta_0$, were estimated when we fitted the model to the household transmission data.

Under the assumption of independent transmission and symptoms, the proportion of transmissions occurring prior to symptom onset (among infectors who develop symptoms) is given by (***Ferretti et al., 2020b***; ***Fraser et al., 2004***)

$$\int_0^\infty f(\tau)\left(1 - F_{inc}(\tau)\right)\mathrm{d}\tau,$$

where $F_{inc}$ is the cumulative distribution function of the incubation period (this was assumed to be known; the exact incubation period distribution we used is given under 'Parameter estimation' below).

## Mechanistic model

Under the mechanistic model (***Hart et al., 2021***), infectors who develop symptoms progress through independent latent ($E$), presymptomatic infectious ($P$) and symptomatic infectious ($I$) stages of infection. We assumed the duration of each stage to be gamma distributed, and infectiousness was assumed to be constant during each stage. Under these assumptions, an expression can be derived for the expected infectiousness, $\beta(\tau \mid \tau_{inc})$, of a host (who develops symptoms) at each time since infection $\tau$, conditional on their incubation period $\tau_{inc}$. We assumed that entirely asymptomatic infected hosts follow the same stage progression as those who develop symptoms, although in this case the distinction between the $P$ and $I$ stages has no epidemiological meaning. Details of the mechanistic approach, including the formula for $\beta(\tau \mid \tau_{inc})$, are provided in the Appendix.

When we fitted this model to the household transmission data, three model parameters were estimated in addition to $\beta_0$. These parameters correspond to:

- The ratio between the mean latent (*E*) period and the mean incubation (combined *E* and *P*) period (where the latter was assumed to be known).
- The mean symptomatic infectious (*I*) period.
- The ratio between the transmission rates when potential infectors are in the *P* and *I* stages.

## Likelihood function

Here, we consider a household of size $n$, in which $n_I$ household members become infected (of whom $n_S$ develop symptoms and $n_A$ remain asymptomatic throughout infection) and $n_U = n - n_I$ remain uninfected. We derive an expression for the likelihood of the parameters of either model of infectiousness, given the entire sequence of infection times of individuals in the household ($t_1 < \ldots < t_{n_I}$) as well as the precise symptom onset time ($t_{s,j}$) of each host, $j$, who develops symptoms. In the case of the mechanistic model, the likelihood also depends on the times at which entirely asymptomatic infected hosts enter the *I* stage of infection (these times are also denoted by $t_{s,j}$, although for asymptomatic infected individuals these times have no epidemiological meaning). Since exact infection times were not available within study households, and it was unknown exactly when each symptomatic infected host developed symptoms within their recorded symptom onset date, we used data augmentation MCMC to fit the two models to the UK household transmission data using this likelihood function (see further details below).

When deriving the likelihood, we made several simplifying assumptions:

- The virus is introduced once into the household (i.e. no subsequent infections from the community occur following the infection of the primary case).
- No co-primary cases (we relaxed this assumption in the Appendix, *Figure 1—figure supplement 5* and *Figure 3—figure supplement 6*).
- Potential bias towards more recent introduction of the virus into the household if community prevalence is increasing, or less recent if prevalence is decreasing (*Britton and Scalia Tomba, 2019*; *Ferretti et al., 2020b*; *Lehtinen et al., 2021*), was neglected.

We denote the expected infectiousness of household member $j$, at time $\tau$ since infection, by $\beta_j(\tau)$. For the mechanistic model in which transmission and symptoms are not independent, this infectiousness is conditional on the duration of the incubation period, $t_{s,j} - t_j$, for a host who develops symptoms (the infectiousness is also conditional on $(t_{s,j} - t_j)$ for an entirely asymptomatic infected host, although this interval has no epidemiological meaning for such individuals). The total (instantaneous) force of infection exerted at time $t$ on each susceptible household member is then

$$\lambda(t) = \sum_{j=1}^{n_I} \beta_j(t - t_j),$$

where $\beta_j(t - t_j) = 0$ for $t \leq t_j$, and the cumulative force of infection is

$$\Lambda(t) = \int_{-\infty}^{t} \lambda(s) \, ds.$$

For $k = 2, \ldots, n_I$, conditional on the sequence of infection times up to time $t_k$, the probability that host $k$ becomes infected at time $t_k$ is given by

$$\lambda(t_k) \exp(-\Lambda(t_k)),$$

where $\exp(-\Lambda(t_k))$ represents the probability of host $k$ avoiding infection from household contacts that occurred before their actual time of infection, $t_k$ (*Cauchemez et al., 2004*; *Ferguson et al., 2005*). This factor, which was not included in the likelihood when we previously estimated the generation time using data from infector-infectee transmission pairs (*Hart et al., 2021*), is required here because of the regularity of household contacts. Since household contacts occur frequently, it is necessary to account explicitly for contacts between infected and susceptible individuals that did not lead to transmission. The inclusion of this factor in the likelihood therefore corrects for the regularity of household

contacts to ensure widely applicable generation time estimates (note that this factor is equal to one in the limit of a very small overall household infectiousness parameter, $\beta_0$).

For $k = n_I + 1, \ldots, n$, conditional on the entire sequence of infection times, $t_1, \ldots, t_{n_I}$, the probability of host $k$ never being infected is given by $\exp(-\Lambda(\infty))$. In the case of independent transmission and symptoms, we have

$$\exp(-\Lambda(\infty)) = \exp(-\beta_0 (n_S + \alpha_A n_A)/n),$$

whereas for the mechanistic model, $\exp(-\Lambda(\infty))$ instead depends on the incubation periods of those hosts who develop symptoms, as well as the corresponding time periods for entirely asymptomatic infected hosts (see the Appendix).

The likelihood contribution from the household, $L(\theta)$, where $\theta$ is the vector of unknown model parameters, can therefore be written as

$$L(\theta) = \prod_{k=1}^{n} L_{k,1}(\theta) L_{k,2}(\theta).$$

Here, $L_{k,1}(\theta)$ is the contribution to the likelihood from the transmission, or absence of transmission, to host $k$, that is,

$$L_{k,1}(\theta) = \begin{cases} 1, & \text{for } k = 1; \\ \lambda(t_k) \exp(-\Lambda(t_k)), & \text{for } k = 2, \ldots, n_I; \\ \exp(-\Lambda(\infty)), & \text{for } k = n_I + 1, \ldots, n. \end{cases}$$

$L_{k,2}(\theta)$ is the contribution from the incubation period of host $k$ (where applicable), that is, for the independent transmission and symptoms model,

$$L_{k,2}(\theta) = \begin{cases} f_{inc}(t_{s,k} - t_k), & \text{if host } k \text{ becomes infected and develops symptoms;} \\ 1, & \text{otherwise;} \end{cases}$$

where $f_{inc}$ is the probability density function of the incubation period (this was assumed to be known; the exact incubation period distribution we used is given below). For the mechanistic model, we also have a contribution to the likelihood from the (in this case not epidemiologically meaningful) times $(t_{s,k} - t_k)$ for entirely asymptomatic infected hosts, so that

$$L_{k,2}(\theta) = \begin{cases} f_{inc}(t_{s,k} - t_k), & \text{for } k = 1, \ldots, n_I; \\ 1, & \text{for } k = n_I + 1, \ldots, n. \end{cases}$$

## Parameter estimation

### Incubation period

For the independent transmission and symptoms model, we assumed a lognormal incubation period distribution with mean 5.8 days and standard deviation 3.1 days (**McAloon et al., 2020**). For the mechanistic model, we assumed a gamma distributed incubation period with the same mean and standard deviation; this was for mathematical convenience, since the incubation period could then be decomposed into the sum of independent gamma distributed latent and presymptomatic infectious periods. Results for the independent transmission and symptoms model using a gamma distributed incubation period are shown in **Figure 1—figure supplement 7**, and uncertainty in the exact parameters of the incubation period distribution is accounted for in **Figure 1—figure supplement 8**.

### Parameter fitting procedure

Unknown model parameters were estimated using data augmentation MCMC. The observed data comprised information about whether or not individuals were ever infected and/or displayed symptoms, symptom onset dates, and for some individuals an upper bound on their infection time (corresponding to the date of a positive PCR test). These data were augmented with (estimated) precise

times of infection and symptom onset (where applicable) for each infected host. No prior assumptions were made about the order of transmissions within each household.

Below, we outline the parameter fitting procedure that we used for the independent transmission and symptoms model. The procedure used for the mechanistic model was similar and is described in the Appendix.

Lognormal priors were assumed for fitted model parameters (these parameters were the mean and standard deviation of the generation time distribution, in addition to the overall infectiousness, $\beta_0$). The priors for the mean and standard deviation of the generation time distribution had medians of 5 days and 2 days, respectively (these choices were informed by previous estimates of the SARS-CoV-2 generation time distribution [*Ferretti et al., 2020a*; *Ferretti et al., 2020b*; *Ganyani et al., 2020*]), and were chosen to ensure a prior probability of only 0.025 that these parameters exceeded very high values of 10 days and 7 days, respectively. The exact priors we used are given in *Appendix 1—table 2*.

Here, we denote the vector of model parameters by $\theta$, and the augmented data by

$$t = \left( t^{(1)}, \ldots, t^{(M)} \right),$$

where $t^{(m)}$ represents the augmented data from household $m = 1, \ldots, M$, and $M$ is the total number of households. We write the (overall) likelihood as

$$L(\theta; t) = \prod_{m=1}^{M} L^{(m)} \left( \theta; t^{(m)} \right),$$

where the likelihood contribution, $L^{(m)} \left( \theta; t^{(m)} \right)$, from each household, $m$, was computed as described in the previous section (i.e. all households in the study were assumed to be independent), and we denote the prior density of $\theta$ by $\pi(\theta)$.

In each step of the chain, we carried out (in turn) one of the following:

1. Propose new values for each entry of the vector of model parameters, $\theta$, using independent normal proposal distributions for each parameter (around the corresponding parameter values in the previous step of the chain). Accept the proposed parameters, $\theta_{prop}$, with probability

$$\min \left( \frac{L(\theta_{prop}; t) \, \pi(\theta_{prop})}{L(\theta_{old}; t) \, \pi(\theta_{old})}, 1 \right),$$

where $\theta_{old}$ denotes the vector of parameter values from the previous step of the chain, and where the augmented data, $t$, remain unchanged in this step.

2. Propose new values for the precise symptom onset times of each symptomatic infected host, using independent uniform proposal distributions (within the day of symptom of onset for each host). For each household, $m$, accept the proposed augmented data, $t_{prop}^{(m)}$, from that household with probability

$$\min \left( \frac{L^{(m)} \left( \theta; t_{prop}^{(m)} \right)}{L^{(m)} \left( \theta; t_{old}^{(m)} \right)}, 1 \right),$$

where $t_{old}^{(m)}$ denotes the corresponding augmented data from the previous step of the chain, and where the model parameters, $\theta$, remain unchanged in this step (i.e. proposed times are accepted/rejected independently for each household, according to the likelihood contribution from that household).

3. Propose new values for the infection time of one randomly chosen symptomatic infected host in each household (in households where there was at least one), using independent normal proposal distributions (around the equivalent times in the previous step of the chain). For each household, $m$, accept the proposed augmented data, $t_{prop}^{(m)}$, from that household with probability

$$\min\left(\frac{L^{(m)}\left(\theta;\boldsymbol{t}_{prop}^{(m)}\right)}{L^{(m)}\left(\theta;\boldsymbol{t}_{old}^{(m)}\right)},1\right).$$

4. Propose new values for the infection time of one randomly chosen asymptomatic infected host in each household (in households where there was at least one), using independent normal proposal distributions (around the equivalent times in the previous step of the chain). For each household, $m$, accept the proposed augmented data, $\boldsymbol{t}_{prop}^{(m)}$, from that household with probability

$$\min\left(\frac{L^{(m)}\left(\theta;\boldsymbol{t}_{prop}^{(m)}\right)}{L^{(m)}\left(\theta;\boldsymbol{t}_{old}^{(m)}\right)},1\right).$$

The chain was run for 10,000,000 iterations; the first 2,000,000 iterations were discarded as burn-in. Posteriors were obtained by recording every 100 iterations of the chain.

## Governance statement

The household study was approved by the PHE Research Ethics and Governance Group as part of the portfolio of PHE's enhanced surveillance activities in response to the pandemic.

## Acknowledgements

Thanks to Pauline Waight, who managed the data for the household study, and to the PHE staff who collected the data and tested the PCR and serum samples. Thanks also to Rob Challen, Julia Gog, Matt Keeling and other members of the Juniper Consortium (https://maths.org/juniper/) for helpful comments about this research.

## Additional information

### Competing interests

Akira Endo: received a research grant from Taisho Pharmaceutical Co., Ltd. The other authors declare that no competing interests exist.

### Funding

| Funder | Grant reference number | Author |
| --- | --- | --- |
| Engineering and Physical Sciences Research Council | EP/R513295/1 | William S Hart |
| National Institute for Health Research | NIHR200929 | Elizabeth Miller |
| Taisho Pharmaceutical Co., Ltd | Research grant | Akira Endo |
| UKRI | EP/V053507/1 | Robin N Thompson |

The funders had no role in study design, data collection and interpretation, or the decision to submit the work for publication.

### Author contributions

William S Hart, Conceptualization, Formal analysis, Investigation, Methodology, Software, Validation, Visualization, Writing – original draft, Writing – review and editing; Sam Abbott, Joel Hellewell, Methodology, Writing – review and editing; Akira Endo, Writing – review and editing; Elizabeth Miller, Nick Andrews, Data curation, Writing – review and editing; Philip K Maini, Methodology, Supervision, Writing – review and editing; Sebastian Funk, Conceptualization, Methodology, Project administration, Supervision, Writing – review and editing; Robin N Thompson, Conceptualization, Methodology, Supervision, Writing – review and editing

## Author ORCIDs

William S Hart http://orcid.org/0000-0002-2504-6860
Akira Endo http://orcid.org/0000-0001-6377-7296
Elizabeth Miller http://orcid.org/0000-0002-1884-0097
Philip K Maini http://orcid.org/0000-0002-0146-9164
Sebastian Funk http://orcid.org/0000-0002-2842-3406
Robin N Thompson http://orcid.org/0000-0001-8545-5212

## Decision letter and Author response

Decision letter https://doi.org/10.7554/eLife.70767.sa1
Author response https://doi.org/10.7554/eLife.70767.sa2

---

# Additional files

## Supplementary files

• Transparent reporting form

## Data availability

All data generated or analysed during this study are included in the manuscript and its supporting files; a Source Data file has been provided for Figure 1. Code for reproducing our results is available at https://github.com/will-s-hart/UK-generation-times (copy archived at swh:1:rev:729266e972315ba3344da430d5de58123fce4e4e).

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

# Appendix 1

## Details of mechanistic model

In this model, each infected host (who develops symptoms) progresses through independent latent ($E$), presymptomatic infectious ($P$) and symptomatic infectious ($I$) stages of infection. The infectiousness of the host during the $P$ and $I$ stages is denoted by $\beta_P$ and $\beta_I$, respectively, and we denote the ratio $\alpha_P = \beta_P/\beta_I$. We assumed the duration of each stage, denoted $y_{E/P/I}$, to be gamma distributed:

$$y_E \sim \mathrm{Gamma}\left(k_E, 1/(k_{inc}\gamma)\right),$$

$$y_P \sim \mathrm{Gamma}\left(k_P, 1/(k_{inc}\gamma)\right),$$

$$y_I \sim \mathrm{Gamma}\left(k_I, 1/(k_I\mu)\right),$$

where we write $X \sim \mathrm{Gamma}(a, b)$ for a gamma distributed random variable with shape parameter $a$ and scale parameter $b$. We assumed that $k_E + k_P = k_{inc}$, so that the incubation period, $\tau_{inc} = y_E + y_P$, is gamma distributed, with

$$\tau_{inc} \sim \mathrm{Gamma}\left(k_{inc}, 1/(k_{inc}\gamma)\right).$$

We fixed the values of the parameters $k_{inc}$ and $1/\gamma$ (which represent the shape parameter of the incubation period distribution and the mean incubation period, respectively) in order to obtain the specified incubation period distribution (the exact values that we assumed are given in *Appendix 1—table 1*). For simplicity, we also assumed that $k_I = 1$, so the symptomatic infectious period is exponentially distributed. The parameters $k_E$ (the shape parameter of the latent ($E$) period distribution), $1/\mu$ (the mean symptomatic infectious ($I$) period) and $\alpha_P$ (the ratio between the transmission rates of hosts in the $P$ and $I$ stages) were estimated when we fitted the model to the household transmission data.

Hosts who remain asymptomatic throughout infection were assumed to follow the same $E/P/I$ stages, although in this case the distinction between the $P$ and $I$ stages has no epidemiological meaning. Stage durations, as well as the value of $\alpha_P$, were assumed to be identical for entirely asymptomatic hosts and those who develop symptoms, so that the generation time distribution is the same for all infected hosts.

### Conditional infectiousness

For a host who develops symptoms, conditional on incubation period $\tau_{inc}$, the expected infectiousness at time since infection $\tau$ is (*Hart et al., 2021*)

$$\beta(\tau \mid \tau_{inc}) = \begin{cases} \alpha_P C\left(\beta_0/n\right)\left(1 - F_{Beta}\left(1 - \tau/\tau_{inc}; k_P, k_E\right)\right), & 0 < \tau < \tau_{inc}, \\ C\left(\beta_0/n\right)\left(1 - F_I\left(\tau - \tau_{inc}\right)\right), & \tau > \tau_{inc}. \end{cases}$$

Here, $\beta_0$ is the overall infectiousness parameter (see Methods in the main text), $n$ is the household size, $F_I(y_I)$ is the cumulative distribution function of the duration of the $I$ stage, $F_{Beta}(x; a, b)$ is the cumulative distribution function of a beta distributed random variable with shape parameters $a$ and $b$, and

$$C = \frac{k_{inc}\gamma\mu}{\alpha_P k_P \mu + k_{inc}\gamma}.$$

The cumulative conditional infectiousness can therefore be calculated to be

$$B(\tau \mid \tau_{inc}) = \int_0^\tau \beta\left(\widetilde{\tau} \mid \tau_{inc}\right)\mathrm{d}\widetilde{\tau}$$

$$= \begin{cases} (\tau - \tau_{inc})\beta(\tau \mid \tau_{inc}) + \frac{\alpha_P C\beta_0}{n}\left[\frac{k_P\tau_{inc}}{k_{inc}}\left(1 - F_{Beta}\left(1 - \tau/\tau_{inc}; k_P + 1, k_E\right)\right)\right], & 0 \leq \tau < \tau_{inc}, \\ (\tau - \tau_{inc})\beta(\tau \mid \tau_{inc}) + \frac{C\beta_0}{n}\left[\frac{\alpha k_P\tau_{inc}}{k_{inc}} + \frac{1}{\mu}F_{Gamma}\left(\tau - \tau_{inc}; k_I + 1, \frac{1}{k_I\mu}\right)\right], & \tau \geq \tau_{inc}, \end{cases}$$

where $F_{Gamma}(x; a, b)$ is the cumulative distribution of a gamma distributed random variable with shape parameter $a$ and scale parameter $b$. The total force of infection exerted on each household member (over the course of infection) is then

$$B\left(\infty \mid \tau_{inc}\right) = \frac{\beta_0}{n}\left(\frac{\alpha_P k_P \gamma \mu \tau_{inc} + k_{inc}\gamma}{\alpha_P k_P \mu + k_{inc}\gamma}\right).$$

The mean of this expression over the incubation period distribution is $\beta_0/n$.

For a host who remains asymptomatic throughout infection, conditional on the combined duration of the $E$ and $P$ stages, $\tau_{inc} = y_E + y_P$, the infectiousness, $\beta\left(\tau \mid \tau_{inc}\right)$, is given by $\alpha_A$ times the corresponding expression for a host who develops symptoms. We note that in this case, $\tau_{inc}$ has no epidemiological interpretation, but this conditional infectiousness was useful when fitting the model to data (see 'Parameter estimation' below).

## Generation time distribution

The generation time, $\tau_{gen}$, for an individual transmission can be written as

$$\tau_{gen} = y_E + y^*,$$

where $y_E$ is the length of the latent ($E$) stage, and $y^*$ is the time from the start of the presymptomatic infectious ($P$) stage to the transmission occurring. As shown by *Hart et al., 2021*, if the effect of susceptible depletion during infection is neglected, $y^*$ has density,

$$f^*\left(y^*\right) = C\left(\alpha_P\left(1 - F_P(y^*)\right) + \int_0^{y^*}\left(1 - F_I(y^* - y_P)\right)f_P\left(y_P\right)\mathrm{d}y_P\right).$$

Using this density, it can be shown that the moments of this distribution are

$$\mathbb{E}\left[\left(y^*\right)^m\right] = \frac{C}{m+1}\left(\alpha_P\mathbb{E}\left[y_P^{m+1}\right] + \mathbb{E}\left[\left(y_P + y_I\right)^{m+1} - y_P^{m+1}\right]\right).$$

In particular,

$$\mathbb{E}\left[y^*\right] = \frac{C}{2}\left(\alpha_P\mathbb{E}\left[y_P^2\right] + 2\mathbb{E}\left[y_P\right]\mathbb{E}\left[y_I\right] + \mathbb{E}\left[y_I^2\right]\right),$$

and

$$\mathrm{Var}\left[y^*\right] = \frac{C}{3}\left(\alpha_P\mathbb{E}\left[y_P^3\right] + 3\mathbb{E}\left[y_P^2\right]\mathbb{E}\left[y_I\right] + 3\mathbb{E}\left[y_P\right]\mathbb{E}\left[y_I^2\right] + \mathbb{E}\left[y_I^3\right]\right) - \left(\mathbb{E}\left[y^*\right]\right)^2.$$

Note that for a gamma distributed random variable, $X \sim \mathrm{Gamma}\left(a, b\right)$, we have

$$\mathbb{E}\left[X^m\right] = \frac{\Gamma\left(a + m\right)}{\Gamma\left(a\right)}b^m = a\left(a + 1\right)\ldots\left(a + \left(m - 1\right)\right)b^m.$$

Therefore, for gamma distributed stage durations, explicit expressions can be obtained for the mean and variance of the generation time distribution,

$$\mathbb{E}\left[\tau_{gen}\right] = \mathbb{E}\left[y_E\right] + \mathbb{E}\left[y^*\right],$$
$$\mathrm{Var}\left[\tau_{gen}\right] = \mathrm{Var}\left[y_E\right] + \mathrm{Var}\left[y^*\right],$$

where the last equality holds because $y_E$ and $y^*$ are assumed to be independent.

## Proportion of presymptomatic transmissions

Among infectors who develop symptoms, the proportion of transmissions occurring prior to symptom onset (neglecting the effect of susceptible depletion during infection) is given by (*Hart et al., 2021*)

$$q_P = \frac{\left(\frac{\beta_P k_P}{k_{inc}\gamma}\right)}{\left(\frac{\beta_P k_P}{k_{inc}\gamma} + \frac{\beta_I}{\mu}\right)} = \frac{\alpha_P k_P \mu}{\alpha_P k_P \mu + k_{inc}\gamma}.$$

## Parameter estimation

The vector of model parameters,

$$\theta = \left( k_E/k_{inc}, 1/\mu, \alpha_P, \beta_0 \right),$$

was estimated by fitting the mechanistic model to the household transmission data.

We assumed independent prior distributions for each entry of $\theta$. Lognormal priors were assumed for $1/\mu$, $\alpha_P$ and $\beta_0$. Since $\alpha_P$ represents the ratio between the transmission rates of hosts in the $P$ and $I$ stages, a prior with median one was used to ensure equal prior probabilities of values above and below one. This prior was also chosen to limit the prior probability of extreme values, with a prior 95% credible interval of [0.2,5]. A beta prior was used for $k_E/k_{inc}$ (which was constrained to lie between 0 and 1), and was chosen to restrict the prior probability of values very close to either 0 or 1. The exact priors we used are given in *Appendix 1—table 3*.

A slightly amended version of the parameter fitting algorithm described in the main text for the independent transmission and symptoms model was used. In particular, we augmented the observed data with:

i.   The infection time, $t_j$, of each infected host.
ii.  The time, $t_{s,j}$, at which each infected host transitioned from the $P$ to $I$ stage.

Note that for hosts who develop symptoms, the time of entry into the $I$ stage corresponds to the symptom onset time. The data were also augmented with this transition time for entirely asymptomatic infected hosts because the conditional infectiousness, $\beta(\tau \mid t_{s,j} - t_j)$, is more straightforward to calculate than $\beta(\tau)$.

In each step of the chain, we carried out (in turn) one of the following:

1. Propose new values for each entry of the vector of model parameters, $\theta$, using a multivariate normal proposal distribution (around the value of $\theta$ in the previous step of the chain; a correlation of 0.5 was used between the proposal distributions of $k_E/k_{inc}$ and $\alpha_P$, and between those of $1/\mu$ and $\alpha_P$). Accept the proposed parameters, $\theta_{prop}$, with probability

$$\min \left( \frac{L(\theta_{prop}; \boldsymbol{t}) \, \pi(\theta_{prop})}{L(\theta_{old}; \boldsymbol{t}) \, \pi(\theta_{old})}, 1 \right),$$

   where $\theta_{old}$ denotes the vector of parameter values from the previous step of the chain, and where the augmented data, $\boldsymbol{t}$ remain unchanged in this step.

2. Propose new values for the precise symptom onset times of each symptomatic infected host, using independent uniform proposal distributions (within the day of symptom of onset for each host). For each household, $m$, accept the proposed augmented data, $\boldsymbol{t}_{prop}^{(m)}$, from that household with probability

$$\min \left( \frac{L^{(m)}\left( \theta; \boldsymbol{t}_{prop}^{(m)} \right)}{L^{(m)}\left( \theta; \boldsymbol{t}_{old}^{(m)} \right)}, 1 \right),$$

   where $\boldsymbol{t}_{old}^{(m)}$ denotes the corresponding augmented data from the previous step of the chain, and where the model parameters, $\theta$, remain unchanged in this step (i.e. proposed times are accepted/rejected independently for each household, according to the likelihood contribution from that household).

3. Propose new values for the infection time of one randomly chosen infected host in each household (either symptomatic or asymptomatic), using independent normal proposal distributions (around the equivalent times in the previous step of the chain). For each household, $m$, accept the proposed augmented data, $\boldsymbol{t}_{prop}^{(m)}$, from that household with probability

$$\min\left(\frac{L^{(m)}\left(\theta;\boldsymbol{t}_{prop}^{(m)}\right)}{L^{(m)}\left(\theta;\boldsymbol{t}_{old}^{(m)}\right)},1\right).$$

4. Propose new values for both the infection time, $t$, and the time of the start of the $I$ stage, $t_s$, holding $(t_s - t)$ constant, for one randomly chosen asymptomatic infected host in each household (in households where there was at least one), using independent normal proposal distributions (around the equivalent times in the previous step of the chain). For each household, $m$, accept the proposed augmented data, $\boldsymbol{t}_{prop}^{(m)}$, from that household with probability

$$\min\left(\frac{L^{(m)}\left(\theta;\boldsymbol{t}_{prop}^{(m)}\right)}{L^{(m)}\left(\theta;\boldsymbol{t}_{old}^{(m)}\right)},1\right).$$

## Relationship between generation time, TOST and serial interval

Here, we consider a randomly chosen infector-infectee pair (in which both the infector and the infectee develop symptoms) within a large, well-mixed population, of which only a small proportion is infected. In that setting, the observed generation time distribution is equal to the normalised infectiousness profile, which will not be true within a household (compare *Figure 1* and *Figure 1— figure supplement 4*). We define:

$$
\begin{aligned}
\tau_{inc,1} &= \text{(incubation period of the infector)}, \\
\tau_{inc,2} &= \text{(incubation period of the infectee)}, \\
\tau_{gen} &= \text{(generation time)}, \\
x_{tost} &= \text{(time from onset of symptoms (of infector) to transmission (TOST))}, \\
x_{ser} &= \text{(serial interval)},
\end{aligned}
$$

where we use $\tau$ for time intervals relative to the time of infection and $x$ for those relative to the time of symptom onset. We denote the probability density functions of these time periods by $f_{inc,1}$, $f_{inc,2}$, $f_{gen}$, $f_{tost}$ and $f_{ser}$, respectively. Note that

$$x_{tost} = \tau_{gen} - \tau_{inc,1},$$

and

$$x_{ser} = x_{tost} + \tau_{inc,2},$$

so that

$$x_{ser} = \tau_{gen} + \tau_{inc,2} - \tau_{inc,1}.$$

In the independent transmission and symptoms model, $\tau_{gen}$ and $\tau_{inc,1}$ are assumed to be independent, and the incubation periods of the infector and infectee are assumed to be drawn independently from the population incubation period distribution, $f_{inc} = f_{inc,1} = f_{inc,2}$. Therefore, the TOST distribution is given by the convolution

$$f_{tost}\left(x_{tost}\right) = \int_0^\infty f_{gen}\left(x_{tost} + \tau\right) f_{inc}\left(\tau\right) \, \mathrm{d}\tau. \tag{1}$$

Assuming that $x_{tost}$ and $\tau_{inc,2}$ are independent, the serial interval distribution can be calculated from the TOST distribution as

$$f_{ser}\left(x_{ser}\right) = \int_0^\infty f_{tost}\left(x_{ser} - \tau\right) f_{inc}\left(\tau\right) \, \mathrm{d}\tau. \tag{2}$$

Note that

$$\mathbb{E}\left[x_{ser}\right] = \mathbb{E}\left[\tau_{gen}\right] + \mathbb{E}\left[\tau_{inc,2}\right] - \mathbb{E}\left[\tau_{inc,1}\right] = \mathbb{E}\left[\tau_{gen}\right],$$

i.e. the generation time and serial interval distributions have the same mean.

For the mechanistic model, we still have $f_{inc,2} = f_{inc}$, and the serial interval distribution can be calculated from the TOST distribution using **Equation 2**. On the other hand, $\tau_{gen}$ and $\tau_{inc,1}$ are not independent, so **Equation 1** connecting the TOST and generation time distributions for the independent transmission and symptoms model does not hold for the mechanistic model. As shown by **Hart et al., 2021**, the TOST distribution for the mechanistic model is, instead, given by

$$f_{tost}(x_{tost}) = \begin{cases} \alpha_P C \left(1 - F_P(-x_{tost})\right), & x_{tost} < 0, \\ C \left(1 - F_I(x_{tost})\right), & x_{tost} \geq 0. \end{cases}$$

Further, under the mechanistic model, the expected number of presymptomatic transmissions generated by an infected host is dependent on their incubation period. As a result, the infector's incubation period does not follow the same distribution as that of the infectee. In particular, by Bayes' theorem, we have

$$f_{inc,1}(\tau_{inc,1}) = p(\tau_{inc,1} \mid 1 \to 2) = \frac{p(1 \to 2 \mid \tau_{inc,1}) f_{inc}(\tau_{inc,1})}{p(1 \to 2)},$$

where we write $1 \to 2$ to denote the occurrence of the transmission from the infector to the infectee. Because we are here considering a large population, the probability of the transmission occurring is proportional to the overall infectiousness of the infector (integrated over the course of infection), $B(\infty)$, so we have

$$f_{inc,1}(\tau_{inc,1}) = \frac{B(\infty \mid \tau_{inc,1}) f_{inc}(\tau_{inc,1})}{B(\infty)} = \left( \frac{\alpha_P k_P \gamma \mu \tau_{inc,1} + k_{inc} \gamma}{\alpha_P k_P \mu + k_{inc} \gamma} \right) f_{inc}(\tau_{inc,1}).$$

The expected incubation period of the infector is then

$$\mathbb{E}[\tau_{inc,1}] = \frac{1}{\gamma} + \frac{\alpha_P k_P \mu}{k_{inc} \gamma (\alpha_P k_P \mu + k_{inc} \gamma)} = \mathbb{E}[\tau_{inc,2}] + \frac{q_P}{k_{inc} \gamma},$$

where $q_P$ is the proportion of transmissions occurring prior to symptom onset.

As a result of the above, the expected values of the generation time and serial interval in the mechanistic model are not equal. Instead, we have

$$\mathbb{E}[x_{ser}] = \mathbb{E}[\tau_{gen}] - \frac{q_P}{k_{inc} \gamma}.$$

Under the values of $k_{inc}$ and $\gamma$ that we assumed (**Appendix 1—table 1**), this gives a mean generation time that is approximately $(1.6 \times q_P)$ days longer than the mean serial interval.

## Extension of framework to account for co-primary cases

In most of our analyses, we assumed that each household transmission chain was initiated by a single primary case, so that all other infected household members were infected from within the household. However, we also relaxed this assumption by extending our framework to account for the possibility of co-primary cases (**Figure 1—figure supplement 5** and **Figure 3—figure supplement 6**). Rather than assuming that all co-primary cases were infected at exactly the same time, we instead assumed that each household member could be infected at any time during a primary infection event that was taken to last one day (the choice of one day was arbitrary but in principle any duration could be used). This enabled us to easily incorporate the possibility of co-primary cases into our data augmentation MCMC approach by adapting the likelihood function as described below.

As in Methods, we here consider a household (of size $n$) in which $n_I$ household members become infected (of whom $n_S$ develop symptoms and $n_A$ remain asymptomatic throughout infection) and $n_U$ remain uninfected. Under either the independent transmission and symptoms model or the mechanistic model, we now denote the total force of infection exerted on each susceptible member of the household by other household members at time $t$ by $\lambda_h(t)$, and the cumulative force of

infection by $\Lambda_h(t)$ (i.e. these correspond to the quantities denoted by $\lambda(t)$ and $\Lambda(t)$, respectively, in Methods). Assuming each (susceptible) household member is also subject to a constant force of infection, $\beta_p$, during a primary event taking place between times $t_{p,\,start}$ and $t_{p,\,end}$, the total force of infection exerted on each susceptible household member at time $t$ is

$$\lambda(t) = \lambda_p(t) + \lambda_h(t),$$

where

$$\lambda_p(t) = \begin{cases} \beta_p, & t_{p,\,start} \le t \le t_{p,\,end}; \\ 0, & \text{otherwise.} \end{cases}$$

The cumulative force of infection is

$$\Lambda(t) = \Lambda_p(t) + \Lambda_h(t),$$

where

$$\Lambda_p(t) = \int_{-\infty}^{t} \lambda_p(s)\,\mathrm{d}s = \frac{\beta_p}{2}\left(t_{p,\,end} - t_{p,\,start} + |t - t_{p,\,start}| - |t_{p,end} - t|\right).$$

We took $t_{p,\,start}$ and $t_{p,\,end}$ to be the start and end of the day of the first household member becoming infected, respectively.

The likelihood contribution from the household, $L(\theta)$, where $\theta$ is the vector of unknown model parameters, is then given by

$$L(\theta) = \frac{1}{1 - \exp(-n\beta_p \times (t_{p,\,end} - t_{p,\,start}))}\prod_{k=1}^{n} L_{k,1}(\theta)\, L_{k,2}(\theta).$$

Here,

$$L_{k,1}(\theta) = \begin{cases} \lambda(t_k)\exp(-\Lambda(t_k)), & \text{for } k = 1,\ldots,n_I; \\ \exp(-\Lambda(\infty)), & \text{for } k = n_I + 1,\ldots,n; \end{cases}$$

and for the independent transmission and symptoms model,

$$L_{k,2}(\theta) = \begin{cases} f_{inc}(t_{s,k} - t_k), & \text{if host } k \text{ becomes infected develops symptoms;} \\ 1, & \text{otherwise;} \end{cases}$$

where $f_{inc}$ is the probability density function of the incubation period, while for the mechanistic model,

$$L_{k,2}(\theta) = \begin{cases} f_{inc}(t_{s,k} - t_k) & \text{for } k = 1,\ldots,n_1; \\ 1 & \text{for } k = n_I + 1,\ldots,n. \end{cases}$$

The factor

$$\frac{1}{1 - \exp(-n\beta_p \times (t_{p,\,end} - t_{p,\,start}))},$$

is included to condition on at least one household member becoming infected during the primary transmission event.

Using this likelihood function, we fitted both models to the household data using the same data augmentation MCMC approach described for the independent transmission and symptoms model in Methods and for the mechanistic model earlier in the Appendix. Alongside other model

parameters, we estimated the probability of each household member becoming infected during the primary transmission event,

$$1 - \exp\left(-\beta_p \times \left(t_{p,\,end} - t_{p,\,start}\right)\right),$$

in the MCMC procedure (in the case we considered, $\left(t_{p,\,end} - t_{p,\,start}\right)$ was always equal to one day, so $\beta_p$ could be calculated from this probability). A uniform prior was assumed for the probability of primary infection.

## Supplementary tables

**Appendix 1—table 1.** Assumed (not fitted) parameter values used for the two models that we considered.

| Parameter | Model | Interpretation | Value | Justification |
|---|---|---|---|---|
| $\alpha_A$ | Both | Relative infectiousness of entirely asymptomatic hosts | 0.35 | Taken from **Buitrago-Garcia et al., 2020** (other values considered in sensitivity analyses) |
| Mean of natural logarithm of the incubation period | Independent transmission and symptoms | Parameter of lognormal incubation period distribution | 1.63 log(day) | Taken from **McAloon et al., 2020** (uncertainty in this value considered in sensitivity analyses) |
| Standard deviation of natural logarithm of the incubation period | Independent transmission and symptoms | Parameter of lognormal incubation period distribution | 0.50 log(day) | Taken from **McAloon et al., 2020** (uncertainty in this value considered in sensitivity analyses) |
| $k_{inc}$ | Mechanistic | Shape parameter of gamma incubation period distribution | 3.5 | Consistent with mean and standard deviation from **McAloon et al., 2020** |
| $1/\gamma$ | Mechanistic | Mean incubation period | 5.8 days | Consistent with mean and standard deviation from **McAloon et al., 2020** |
| $k_I$ | Mechanistic | Shape parameter of (gamma) symptomatic infectious period distribution | 1 | Assumed |

**Appendix 1—table 2.** Fitted parameters in the independent transmission and symptoms model, the prior distributions used, and the posterior means and 95% credible intervals obtained.

| Parameter | Prior | Posterior mean (95% CrI) |
|---|---|---|
| Mean generation time | Lognormal(1.6,0.35) [prior median 5.0 days, 95% CrI 2.5–9.8 days] | 4.2 days (3.3–5.3 days) |
| Standard deviation of generation time distribution | Lognormal(0.7,0.65) [prior median 2.0 days, 95% CrI 0.6–7.2 days] | 4.9 days (3.0–8.3 days) |
| Overall infectiousness parameter, $\beta_0$ | Lognormal(0.7,0.8) [prior median 2.0, 95% CrI 0.4–9.7] | 1.7 (1.4–1.9) |

**Appendix 1—table 3.** Fitted parameters in the mechanistic model, the prior distributions used, and the posterior means and 95% credible intervals obtained.

| Parameter | Prior | Posterior mean (95% CrI) |
|---|---|---|
| Ratio of mean durations of the latent (E) and incubation (combined E and P) periods, $k_E/k_{inc}$ | Beta(2.1,2.1) [prior median 0.5, 95% CrI 0.1–0.9] | 0.2 (0.03–0.5) |
| Mean symptomatic infectious (I) period, $1/\mu$ | Lognormal(1.6,0.8) [prior median 5.0 days, 95% CrI 1.0–23.8 days] | 5.0 days (3.2–7.5 days) |
| Ratio of transmission rates in the P and I stages, $\alpha_P$ | Lognormal(0,0.8) [prior median 1.0, 95% CrI 0.2–4.8] | 3.1 (1.2–6.9) |
| Overall infectiousness parameter, $\beta_0$ | Lognormal(0.7,0.8) [prior median 2.0, 95% CrI 0.4–9.7] | 1.8 (1.5–2.1) |

**Appendix 1—table 4.** The means and standard deviations of the generation time, TOST and serial interval distributions shown in *Figure 2*.

Other than the generation time distribution for the independent transmission and symptoms model (which is lognormal with the specified mean and standard deviation), none of the remaining distributions take a simple parametric form.

| Model | Distribution | Mean | Standard deviation |
|---|---|---|---|
| | Generation time | 4.2 days | 4.9 days |
| | TOST | −1.6 days | 5.8 days |
| Independent transmission and symptoms | Serial interval | 4.2 days | 6.6 days |
| | Generation time | 5.9 days | 4.8 days |
| | TOST | −1.1 days | 4.9 days |
| Mechanistic | Serial interval | 4.7 days | 5.8 days |

