## [Editor Report]

This paper is a timely update to the authors previous work and will be of interest to those working on public health responses and the mathematical modelling of infectious diseases. In this work the authors infer the generation interval of SARS–CoV–2 which can allow for the assessment of public health measures. The derivation of the likelihood function is also of interest to mathematical modellers as it allows for the inference of the generation interval from data sets where susceptible depletion may dominate infection dynamics.

---

## [Decision Letter]

**Decision letter after peer review:**

[Editors’ note: the authors submitted for reconsideration following the decision after peer review. What follows is the decision letter after the first round of review.]

Thank you for submitting the paper "Inference of SARS–CoV–2 generation times using UK household data" for consideration by *eLife*. Your article has been reviewed by 3 peer reviewers, and the evaluation has been overseen by a Reviewing Editor and a Senior Editor. The following individuals involved in review of your submission have agreed to reveal their identity: Rowland Raymond Kao (Reviewer #1); Eamon Conway (Reviewer #2).

We are sorry to say that, after consultation with the reviewers, we have decided that this work will not be considered further for publication by *eLife*.

Specifically, all of the reviewers agreed that there wasn't enough novelty in the manuscript, given that the main methodology has been previously published, to be considered in *eLife*. There were also concerns over the generalisability of the work. The work is very well written and important but would be better suited in a more specialised journal. The authors should consider emphasising the changes to the likelihood function to deal with household data, since this is a novel contribution of the work.

*Reviewer #1:*

This paper extends a previous analytical method that the authors developed to evaluate the time to infectiousness of COVID–19, in order to evaluate differences in the generation interval across different time periods during the course of the pandemic in England in 2020. The time to infectiousness (i.e. how long is it until infected individuals start producing virus in a way that is a risk of infecting others) is a generalisable concept. That is unless we expect there to be inherent differences in the way infected individuals progress to becoming infectious (when looking at distributions of outcomes, comparing between populations of interest) we can take a result from one population of individuals, and assume that it gives us a reasonable idea of how long it takes to become infectious, in another population. Differences in the way people come into contact with each other will have some influence on this, but generally speaking if a person is infectious after 4 days in China, you should be considering a person to be a risk of infecting others after 4 days in other countries as well.

In contrast, generation time (how long does it take an infected person, on average, to infect the persons they are going to infect?) depends strongly not just on the inherent characteristics of the virus, and progression of disease in individuals, but also (more strongly that time to infectiousness) the circumstances of contact between individuals. Because generation time is tied to so many other factors, one of the most reliable ways to estimate generation times is to analyse data where there are groups of in–contact individuals where there is likely to be highly likely that there is only one generation of transmission involved (where contacts between individuals are clustered, possibly two but with three generations highly unlikely). In this case, the most important unknowns are the time from when individuals are infected to when become infectious and the time to when they test positive – the requirement for time to infectiousness is why the methods used in the initial paper are appropriate for generating better generation time estimates.

As most published results relate to the very early stages of the pandemic in China where extensive contact tracing was done, there is some interest in understanding whether the generation times differ substantially in other locations and if they change over time (and therefore, why). In this analysis, Hart et al. estimate generation times across three, three month time periods using household contact data in England in 2020, and show differences in generation time estimates depending on the method used (in particular, when considering an approach which ties infectiousness to symptomatic development which they showed provided better results compared to other methods in their previous paper) and the period of 2020 over which the estimates are taken. While the result appears technically robust for the data analysed, its usefulness is limited by difficulty in extending the results – while a different dataset from ones used for the analyses in China they refer to, and from the result of Challen et al. that looked at contacts of international travellers in the UK, it is also in its own way quite specific and further breakdown of possible factors would be worthwhile. First, the limitations to household contacts means that it is not representative of general transmission in the population – household contacts are high risk, with many opportunities for transmission and may therefore be relatively short. Generalised contacts outside of households are likely to be less frequent and often of shorter duration and more strongly affected by diurnal and weekly rhythms. Second, it is also known that demographic factors such as ethnicity and income are strongly linked to infection and severe infection risk. While this does not tell us directly about any links to infectiousness and infectious contact, it is reasonable to consider a connection – and therefore a link to generation times. As such, in this relatively small sample (172 households, with much higher numbers in the first 3 months, compared to the middle or last three) differences in demographics may influence generation times as well. Finally, the α variant, first identified in Kent, was probably circulating for much of the final three months of this analysis – dominant by early 2021 in the UK, it would have had a variable proportion across much of those final three months, and also varied geographically in terms of proportion as well, with a much earlier rise in the SE and in London. Unless those proportions are known, it would be difficult to know how much differences in generation times are due to the variant, to demographics, or other, possibly behavioural factors. Thus, some caution should be applied before taking general lessons from it, at least in the absence of those additional considerations.

In my view, the bulk of the methodological innovation was in the original paper and therefore as it stands, the principle interest is in the estimates of the generation times themselves. However, while I do think there is some interest in these results really in my view, they are specific and situational. The data are limited as they are to a relatively small number of households, involving only household contacts, where the uncertainties of variants of concern, and demographics including ethnicity, income, nature of housing, etc. make it difficult to interpret the results with real generality. I would also recommend that the authors include a discussion of the biases that may limit the generality of their work.

*Reviewer #2:*

In this work, Hart et al. infer the generation interval for SARS–CoV–2 using infector–infectee pairs from household data. The generation interval is obtained across three different time intervals (March–April, May–August and September–November) and using both an "independent transmission" model and the "mechanistic" model that was originally proposed in Hart et al. 2021. The main result is that the inferred generation interval in September–November has decreased compared to the earlier months of the pandemic, irrespective of the model considered. Overall, the conclusions drawn in the paper are well supported and have been shown to be robust through a thorough sensitivity analysis.

Strengths

– They use a mechanistic model to account for the change in infectivity at symptom onset.

– A major strength of this investigation is that they can observe the dynamics of the generation time over three different time periods of the pandemic. To my knowledge, this is a novel result that allows for a more up to date understanding of SARS–CoV–2 transmission.

– Whilst not highlighted in the text, it appears that there has been significant effort to extend the likelihood function to appropriately model household dynamics. This is non–trivial work in my opinion, and I believe the details of the derivation will be of use to mathematical modellers that deal with susceptible depletion in their data.

Weaknesses

– The main weakness of the paper in its current form is that the analysis appears superficial, with a large amount of curve fitting and very little explanation. It would be beneficial if the authors delved more deeply into their results, especially with the mechanistic model. It would be very interesting to relate the changes in generation time to mechanisms of transmission.

– The authors calculate the mean and standard deviation of the generation interval across three different time points; however, they only present one figure with the distribution of the generation time (Figure 2). It would be interesting to know how the generation time distribution changes in time, as opposed to just the mean and standard deviation. I believe that such an analysis would link nicely to their previous work, where they highlight the importance of ongoing public health measures such as contact tracing.

I would like to congratulate the authors on a timely update to their work. I thoroughly enjoyed seeing their updated results, especially as some of the questions addressed have been of interest to myself. I do however have some recommendations.

I understand that writing a rather mathematical paper for a general audience can be quite complicated, but I feel in this case that the authors have done themselves a disservice by not emphasising the technical concepts in the paper. At first read it appears that the authors have taken their model and fitted values, which is not particularly interesting. It was only once I made it to the Materials and methods section where I found the significant extension on previous work. I believe highlighting the adaptation of the likelihood function to account for the household level data was non–trivial and should be mentioned earlier (I believe this could be placed in the Results section), adding to the appeal of the paper. I note that susceptible depletion is mentioned in the main text, but I believe you should elaborate on how the likelihood function has been constructed to account for this.

Throughout the work the posterior mean has been used as a point estimate for parameter values. I believe a more natural point estimate would be to choose the maximum of the posterior distribution. I notice that when looking at the posterior distributions of the mechanistic model (Figure S2), the maximum value of the posterior and the posterior mean differ by a wide mark for α_p and k_E/k_inc. The impact of this choice might be minimal, but I believe it should be investigated.

It would be interesting to know how the generation time distribution changes in time, as opposed to just the mean and standard deviation. This would be a simple extension where they take the point estimates for multiple time points to show the temporal variation. I believe that such an analysis would link nicely to your previous work.

I am uncertain why the arguments of the paragraph at line 227 are required. It appears that the point is to justify the inclusion of a 1/n factor in the force of infection, however, I believe this is an obvious factor to include (I would use 1/(n–1) rather than 1/n though) that does not require parameter fitting to understand. If you were to consider a multigroup SIR model with varying population numbers the 1/(n–1), where n is the number of individuals in the group, is included so as the force of infection acts on the proportion of individuals that are susceptible. If this was not the case, then a different β would be required in each group. As you argue that the β value is a constant and does not vary between households it makes sense that the β value must be scaled by the number of individuals in the household, otherwise you would need a different β value for each house (which would be impossible to infer given the small household sizes).

For reproducibility and transparency, I would like the authors to provide all code used to generate results, in line with *eLife*'s policies on availability of data, software and research materials. This will allow other researchers to implement the methods they have developed on other data sets, but also enable confirmation that there is no coding mistakes.

*Reviewer #3:*

The authors have previously published a mechanistic model for inferring infectiousness profile that explicitly models dependence of the risk of onward transmission on the onset of symptoms on an individual. In the present study, they apply this model as well as another more commonly used model which assumes these two things (transmission risk and onset of symptoms) to be independent, to data from a household study conducted from March–Nov 2020 in the UK. Both the models find that the mean generation time in Sept–Nov 2020 is shorter than in the earlier periods of the study.

This is well–presented study with careful analysis and extensive sensitive analysis which shows that the modelled estimates are robust to a range of assumptions.

[Editors’ note: further revisions were suggested prior to acceptance, as described below.]

Thank you for resubmitting your article "Inference of the SARS–CoV–2 generation time using UK household data" for consideration by *eLife*. Your article has been reviewed by 3 peer reviewers, and the evaluation has been overseen by a Reviewing Editor and a Senior Editor. The following individuals involved in review of your submission have agreed to reveal their identity: Rowland Raymond Kao (Reviewer #1); Eamon Conway (Reviewer #2).

This paper is a timely update to the authors previous work and will be of interest to those working on public health responses and the mathematical modelling of infectious diseases. In this work the authors infer the generation interval of SARS–CoV–2 which can allow for the assessment of public health measures. The derivation of the likelihood function is also of interest to mathematical modellers as it allows for the inference of the generation interval from data sets where susceptible depletion may dominate infection dynamics.

As is customary in *eLife*, the reviewers have discussed their critiques with one another. What follows below is the Reviewing Editor's edited compilation of the essential and ancillary points provided by reviewers in their critiques and in their interaction post–review. Please submit a revised version that addresses these concerns directly. Although we expect that you will address these comments in your response letter, we also need to see the corresponding revision clearly marked in the text of the manuscript. Some of the reviewers' comments may seem to be simple queries or challenges that do not prompt revisions to the text. Please keep in mind, however, that readers may have the same perspective as the reviewers. Therefore, it is essential that you attempt to amend or expand the text to clarify the narrative accordingly.

Essential revisions:

1) While the observation of reduced generation times is both useful if true, and potentially plausible, it may not be robust. The overlap between the posterior estimates of generation times etc. are quite broad – and looking across three periods it doesn't seem like it would take much to change the trends in even the mean values.

2) In particular, the size of the study is not that large, and in each household, it seems from the Miller paper, that only two PCR tests were taken – as the approach does not consider the impact of latent processes (i.e. missed infections) it would be important to know whether a slight bias in missed infections across periods would impact on the conclusions.

3) The authors also state (line 573) that "Potential bias towards more recent infection of the primary host if community prevalence is increasing, or less recent if prevalence is decreasing (Britton and Scalia 900 Tomba, 2019; Ferretti et al., 2020b; Lehtinen et al., 2021), was neglected." Could this also provide some possible explanation for the shift in generation times? Especially given that the justify their assumption in part on the analysis across individual months, and there are relatively few recruited households (on the order of 10 I think, based on Figure 3 in the supplement).

4) The authors also say that (line 150) " we corrected for the regularity of household contacts to derive more widely applicable estimates of the generation time. We did this by including a factor in the likelihood to account for each infected individual avoiding infection from household contacts that occurred prior to their actual time of infection (see Materials and Methods for full details of our approach)." This sounds really interesting and would greatly increase the generality of the outcome. But unfortunately, from the description in the material and methods I was not able to figure out exactly why this was – which doesn't mean it’s wrong of course, but it would be helpful to me to have a more detailed description.

5) The authors state that on line 163 that "point estimates for each model using the posterior means of fitted model parameters because the mode of the joint posterior distribution could not easily be calculated from the output of the MCMC

procedure." It would be important to know whether there are any correlations in the parameter posteriors that might make inappropriate.

6) I spent some time trying to understand if there could be any issue causing the higher fraction of pre symptomatic transmission, which is the most unexpected result, and I could not find any obvious one. Same for the high variance of the generation time distribution (though this and the high pre–symptomatic transmission could be related). Hence, I think that these results can be published in the current form.

On the other way, the temporal changes in generation time do not seem to account for the epidemic dynamics and therefore would be biased upward in Spring 2020 and downward in Autumn 2020 as observed. The authors are aware of that as they explain in the Discussion, but I think that the author should either correct for this effect in their approach or clarify better how this effect is accounted for and what may its contribution be.

*Reviewer #1:*

The additional work done by the authors has been considerable and substantially increased the potential value of the work. In particular, the addition of data augmentation MCMC helps to provide greater depth to the outcomes, and the identification of declining generation times useful (especially if it could be established in 'real time' – i.e. rather than retrospectively, but to aid in understanding ongoing epidemics) and interesting.

I do have a few concerns which in my view need to be addressed before it would be suitable for publication in *eLife*.

First, while the observation of reduced generation times is both useful if true, and potentially plausible, it may not be robust. The overlap between the posterior estimates of generation times etc. are quite broad – and looking across three periods it doesn't seem like it would take much to change the trends in even the mean values.

In particular, the size of the study is not that large, and in each household, it seems from the Miller paper, that only two PCR tests were taken – as the approach does not consider the impact of latent processes (i.e. missed infections) it would be important to know whether a slight bias in missed infections across periods would impact on the conclusions.

The authors also state (line 573) that "Potential bias towards more recent infection of the primary host if community prevalence is increasing, or less recent if prevalence is decreasing (Britton and Scalia 900 Tomba, 2019; Ferretti et al., 2020b; Lehtinen et al., 2021), was neglected." Could this also provide some possible explanation for the shift in generation times? Especially given that the justify their assumption in part on the analysis across individual months, and there are relatively few recruited households (on the order of 10 I think, based on Figure 3 in the supplement).

The authors also say that (line 150) " we corrected for the regularity of household contacts to derive more widely applicable estimates of the generation time. We did this by including a factor in the likelihood to account for each infected individual avoiding infection from household contacts that occurred prior to their actual time of infection (see Materials and Methods for full details of our approach)." This sounds really interesting and would greatly increase the generality of the outcome. But unfortunately, from the description in the material and methods I was not able to figure out exactly why this was – which doesn't mean it’s wrong of course, but it would be helpful to me to have a more detailed description.

The authors state that on line 163 that "point estimates for each model using the posterior means of fitted model parameters because the mode of the joint posterior distribution could not easily be calculated from the output of the MCMC

procedure." It would be important to know whether there are any correlations in the parameter posteriors that might make inappropriate.

*Reviewer #2:*

I'd like to thank the authors for updating the manuscript in a very thorough manner, I really enjoyed reading through the revisions. I believe that the authors have addressed all of my concerns.

*Reviewer #4:*

This excellent paper suggests that despite extensive studies, we have not yet reached a full understanding of the generation time of SARS–CoV–2. The study is a robust examination of the subject of generation time within households in UK, which may not be representative of transmission in other contexts. It is unclear to the reviewers if temporal changes in generation time are real and attributable to e.g. the appearance of B.1.177.

This work is sound. While surprising, the results are supported by multiple statistical/modelling approaches and robustness analyses, and believable.

The three most striking results are:

1) The width of the generation time distribution is much wider than in previous works. While this is undoubtedly surprising, the explanation by the authors is believable: home quarantine in the UK is probably less effective in stopping late transmissions within households and may even amplify them.

2) The fraction of pre-symptomatic transmissions is >70%, quite high compared to most previous estimates. Combined with the high number of fully asymptomatic individuals, it would imply that <20% of transmissions come from individuals showing symptoms. This result seems also hard to square with the previous one, which would suggest a wide distribution of TOST. Of course, this estimate may be affected by the setting, since the analysis is restricted to households and therefore a higher force of infection.

3) According to this work, the generation time changed between spring 2020 and autumn 2020 in the UK. This corresponds to the arrival of the B.1.177 lineage, probably more infectious than previous variants, but also to a different epidemiological phase of the epidemic: lockdown followed by gradual reopening in spring/summer, with a corresponding decrease in incidence, then a new wave in autumn with an increase in the number of cases until November. The authors do not correct for this epidemiological dynamic, therefore leaving open the possibility that it would cause an apparent change in generation time similar to the observed one. Other explanations (e.g. behavioural or reporting ones) may be possible.

It is important to remark that many of the results of the mechanistic model may be affected by the assumption that longer incubation intervals correspond to higher infectiousness. The agreement with the results of the simpler model with independent incubation period and generation time implies that this assumption is not relevant for the main results (with the possible exception of the longer mean generation time).

Recommendations:

The results of the paper look really robust.

I spent some time trying to understand if there could be any issue causing the higher fraction of pre symptomatic transmission, which is the most unexpected result, and I could not find any obvious one. Same for the high variance of the generation time distribution (though this and the high pre–symptomatic transmission could be related). Hence, I think that these results can be published in the current form.

On the other way, the temporal changes in generation time do not seem to account for the epidemic dynamics and therefore would be biased upward in Spring 2020 and downward in Autumn 2020 as observed. The authors are aware of that as they explain in the Discussion, but I think that the author should either correct for this effect in their approach or clarify better how this effect is accounted for and what may its contribution be.

---

## [Author Response]

[Editors’ note: The authors appealed the original decision. What follows is the authors’ response to the first round of review.]

Specifically, all of the reviewers agreed that there wasn't enough novelty in the manuscript, given that the main methodology has been previously published, to be considered in eLife. There were also concerns over the generalisability of the work. The work is very well written and important but would be better suited in a more specialised journal. The authors should consider emphasising the changes to the likelihood function to deal with household data, since this is a novel contribution of the work.

Thank you for your helpful feedback and comments that have allowed us to improve our manuscript. As a Research Advance article, the main aim of our study is to update the results of our previous work with more recent estimates of the SARSCoV-2 generation time. However, we agree with Reviewer 2 that the adaptation of the likelihood function to estimate the generation time using household data represents a significant methodological extension of our earlier work.

As recommended by Reviewer 2, we have therefore added a new paragraph to the start of the Results to improve the exposition of this methodological advance (lines 137-144 and 150-154). We describe how we used a data augmentation MCMC approach in which we augmented the observed data with both estimated times of infection and the precise times at which symptomatic infected hosts developed symptoms (compared to our earlier work in which only symptom onset times were imputed; lines 140-142). This allowed us to account (in the likelihood function) for two important differences between the household transmission data considered here and the data from infector-infectee pairs used in our previous study: first, we accounted for uncertainty in exactly who infected whom within a household by summing together likelihood contributions corresponding to infection by different possible infectors (lines 142-144 and 150); second, we corrected for the regularity of household contacts by including a factor in the likelihood function that accounts for each infected individual avoiding infection from household contacts that occurred prior to their actual time of infection (lines 150-154).

In addition, a further novel component of our research compared to other previous studies in which the generation time has been estimated is the inclusion of the contribution of entirely asymptomatic infectors in the likelihood function. We also highlight this clearly in the revised manuscript (lines 613-616).

We hope our responses to the points raised by Reviewer 1 below alleviate the initial concerns about the generalisability of our results. In particular, we emphasise that our approach specifically corrects for the regularity of household contacts to give more widely applicable estimates of the generation time (see lines 150-154, 654-659 and 675-677 of the revised manuscript). Since household data are routinely collected during epidemics, our modelling framework can be used to estimate the generation time (an important measure describing the timescale of realised transmission) during future outbreaks of many different pathogens. Furthermore, our general finding that the generation time changes temporally is important, as it highlights the importance of monitoring the generation time throughout epidemics so that transmission can be characterised accurately. Finally, we emphasise that our results provide some of the most up-to-date estimates of the SARS-CoV-2 generation time. We therefore believe that this research is both generalisable and of widespread interest.

Reviewer #1:This paper extends a previous analytical method that the authors developed to evaluate the time to infectiousness of COVID–19, in order to evaluate differences in the generation interval across different time periods during the course of the pandemic in England in 2020. The time to infectiousness (i.e. how long is it until infected individuals start producing virus in a way that is a risk of infecting others) is a generalisable concept. That is unless we expect there to be inherent differences in the way infected individuals progress to becoming infectious (when looking at distributions of outcomes, comparing between populations of interest) we can take a result from one population of individuals, and assume that it gives us a reasonable idea of how long it takes to become infectious, in another population. Differences in the way people come into contact with each other will have some influence on this, but generally speaking if a person is infectious after 4 days in China, you should be considering a person to be a risk of infecting others after 4 days in other countries as well.In contrast, generation time (how long does it take an infected person, on average, to infect the persons they are going to infect?) depends strongly not just on the inherent characteristics of the virus, and progression of disease in individuals, but also (more strongly that time to infectiousness) the circumstances of contact between individuals. Because generation time is tied to so many other factors, one of the most reliable ways to estimate generation times is to analyse data where there are groups of in–contact individuals where there is likely to be highly likely that there is only one generation of transmission involved (where contacts between individuals are clustered, possibly two but with three generations highly unlikely). In this case, the most important unknowns are the time from when individuals are infected to when become infectious and the time to when they test positive – the requirement for time to infectiousness is why the methods used in the initial paper are appropriate for generating better generation time estimates.

We thank the reviewer for their helpful comments and are pleased that they recognise that our mechanistic model is appropriate for estimating the generation time. The reviewer is correct that the distribution of the time to infectiousness is likely to be more consistent between settings than that of the generation time, which depends on both the infectiousness of infected hosts at different times since infection and on behavioural factors (for example, if infected individuals self-isolate after developing symptoms, this acts to reduce the generation time; adding this explicit link between symptoms and infectiousness was the main advance of our original *eLife* article). Unfortunately, however, in many scenarios it is most important to estimate the generation time (rather than inherent infectiousness), since the generation time describes realised transmission. For example, estimates of the time-dependent reproduction number depend on the generation time distribution, since it is a characteristic of realised transmission in the population. As a result, obtaining up-to-date and location-specific estimates of the SARS-CoV-2 generation time is crucial, particularly in light of our finding that the generation time changes

As most published results relate to the very early stages of the pandemic in China where extensive contact tracing was done, there is some interest in understanding whether the generation times differ substantially in other locations and if they change over time (and therefore, why). In this analysis, Hart et al. estimate generation times across three, three month time periods using household contact data in England in 2020, and show differences in generation time estimates depending on the method used (in particular, when considering an approach which ties infectiousness to symptomatic development which they showed provided better results compared to other methods in their previous paper) and the period of 2020 over which the estimates are taken. While the result appears technically robust for the data analysed, its usefulness is limited by difficulty in extending the results – while a different dataset from ones used for the analyses in China they refer to, and from the result of Challen et al. that looked at contacts of international travellers in the UK, it is also in its own way quite specific and further breakdown of possible factors would be worthwhile.

We agree with the reviewer that investigating whether the generation time varies by location and temporally is an interesting research question. Since, as we show, the generation time actually does vary temporally, it is crucial to monitor the generation time during epidemics and use the most up-to-date estimates when analysing population-level transmission.

While we used data from households in our analyses, our approach corrects for the regularity of household contacts to obtain widely applicable generation time estimates (see lines 150-154, 654-659 and 675-677 of the revised manuscript and our response to the reviewer’s next point below). Since household data are routinely collected, we contend that this manuscript provides a useful advance on our previous manuscript (which considered data from known transmission pairs) by providing a general framework for estimating the generation time, as well as some of the most up-to-date SARS-CoV-2 generation time estimates currently available.

We also agree with the reviewer that a further breakdown of possible factors would be a worthwhile extension of this research. Of course, doing this would require data on the characteristics of individuals and households (e.g. ages or socio-economic statuses of different individuals) to be available. In the Discussion of the revised manuscript, we explain the need to conduct such analyses in future to understand how the generation time depends on specific characteristics more clearly (lines 682-686).

First, the limitations to household contacts means that it is not representative of general transmission in the population – household contacts are high risk, with many opportunities for transmission and may therefore be relatively short. Generalised contacts outside of households are likely to be less frequent and often of shorter duration and more strongly affected by diurnal and weekly rhythms.

We agree that the high frequency of household contacts would be expected to lead to shorter generation times within households than in the wider population. However, we explicitly correct for this in our analysis. In the revised manuscript, we now highlight in both the Results (lines 150-154) and the Discussion (lines 654-657) that we include the regularity of household contacts and the availability of susceptible hosts in households in the likelihood function to derive widely applicable estimates of the generation time. These estimates, which correspond to the generation time assuming a constant supply of susceptibles during infection (lines 227-228, 238-240 and 654-657), can then be conditioned to specific population structures (lines 657-659). For example, we estimated the realised generation times within the study households in Figure 1—figure supplement 4. As expected, these household generation times are shorter than our main estimates in Figure 1 (lines 240-249, 657-659 and 675-677).

Moreover, our work demonstrates the important principle that changes in the generation time can be detected using data from household studies, highlighting both the importance of continued monitoring of the generation time and the role of household data in monitoring efforts (lines 686-692 of the revised manuscript). Finally, we note that household data have previously been used to estimate the generation time for other pathogens – see particularly the highly cited study of influenza by Ferguson et al. (https://doi.org/10.1038/nature04017) to which we refer in our manuscript.

Second, it is also known that demographic factors such as ethnicity and income are strongly linked to infection and severe infection risk. While this does not tell us directly about any links to infectiousness and infectious contact, it is reasonable to consider a connection – and therefore a link to generation times. As such, in this relatively small sample (172 households, with much higher numbers in the first 3 months, compared to the middle or last three) differences in demographics may influence generation times as well.

While we agree with the reviewer that the accuracy of our estimates may have been impacted if the study households were not representative of the wider population, we do not believe this caveat to be any more specific to our study than to other studies in which the SARS-CoV-2 generation time has been estimated. In fact, our sample size is larger than those used in all other such studies of which we are aware. We discuss this point in our revised manuscript (lines 679-682) and note that comparing the generation time between individuals/households of different characteristics is an interesting and important area for future work (lines 682-686).

Finally, the Alpha variant, first identified in Kent, was probably circulating for much of the final three months of this analysis – dominant by early 2021 in the UK, it would have had a variable proportion across much of those final three months, and also varied geographically in terms of proportion as well, with a much earlier rise in the SE and in London. Unless those proportions are known, it would be difficult to know how much differences in generation times are due to the variant, to demographics, or other, possibly behavioural factors. Thus, some caution should be applied before taking general lessons from it, at least in the absence of those additional considerations.

Thank you for this interesting comment. In fact, the Public Health England household study underlying our results included genomic surveillance. The Αlpha variant was only responsible for infections in two study households, so we can be confident that this variant was not responsible for our finding of a temporal decrease in the generation time. Since this is an important point, we have now stated it clearly in both the Results and Discussion of the revised manuscript (lines 338-342, 588-592 and 598-601). If more recent data become available, obtaining further updated generation time estimates in light of novel variants is an important area of future work (as noted in lines 601-603 of the revised submission).

In my view, the bulk of the methodological innovation was in the original paper and therefore as it stands, the principle interest is in the estimates of the generation times themselves.

As far as we understand, the key criterion for publication of a Research Advance manuscript in *eLife* is that it provides new results that build on the original published *eLife* article. We would therefore request that our submission – which builds on our previously published research by providing updated generation time estimates and demonstrates temporal variation in the generation time – is considered in this context.

We would also like to emphasise that the adaptation of our approach to estimate the generation time using household data (rather than data from transmission pairs) required a substantial advance in our methodology. We have improved the exposition of this methodological advance by adding a new paragraph to the Results of our revised manuscript (lines 137-144 and 150-154) as recommended by Reviewer 2.

In our revised submission, we have also furthered our methodological innovation by adding a new analysis in which we relax the assumption that each household infection chain was initiated by a single primary case, instead accounting for the possibility of co-primary infections (Figure 1—figure supplement 5, Figure 3—figure supplement 6, and lines 360-378 and 643-650). The novel way in which we incorporated co-primary cases is described in detail in lines 1684-1741 of the Appendix in our revised submission. Even with this extension to our approach, our main qualitative finding of a temporal decrease in the generation time was unchanged (Figure 3—figure supplement 6 and lines 377-378 and 648-650).

However, while I do think there is some interest in these results really in my view, they are specific and situational. The data are limited as they are to a relatively small number of households, involving only household contacts, where the uncertainties of variants of concern, and demographics including ethnicity, income, nature of housing, etc. make it difficult to interpret the results with real generality. I would also recommend that the authors include a discussion of the biases that may limit the generality of their work.

We hope our responses to the points above reassure the reviewer about the generalisability of our results. The household study analysed here involves a larger number of participants than previous studies, we explicitly account for the repetitiveness of household transmission when deriving widely applicable generation time estimates, and we provide information about variants of concern. We thank the reviewer for their helpful comments, allowing us to make these points more clearly in our revised submission, and – as recommended – we have now included a discussion of the limitations of our study in the revised manuscript (lines 652-659 and 675-692).

Reviewer #2:In this work, Hart et al. infer the generation interval for SARS–CoV–2 using infector–infectee pairs from household data. The generation interval is obtained across three different time intervals (March–April, May–August and September–November) and using both an "independent transmission" model and the "mechanistic" model that was originally proposed in Hart et al. 2021. The main result is that the inferred generation interval in September–November has decreased compared to the earlier months of the pandemic, irrespective of the model considered. Overall, the conclusions drawn in the paper are well supported and have been shown to be robust through a thorough sensitivity analysis.

We thank the reviewer for their useful comments and suggestions and are pleased that the reviewer considers our conclusions to be well supported and robust.

Strengths– They use a mechanistic model to account for the change in infectivity at symptom onset.– A major strength of this investigation is that they can observe the dynamics of the generation time over three different time periods of the pandemic. To my knowledge, this is a novel result that allows for a more up to date understanding of SARS–CoV–2 transmission.– Whilst not highlighted in the text, it appears that there has been significant effort to extend the likelihood function to appropriately model household dynamics. This is non–trivial work in my opinion, and I believe the details of the derivation will be of use to mathematical modellers that deal with susceptible depletion in their data.

We thank the reviewer for highlighting some of the key strengths of our study. We agree that the methodological advance in this study is important and useful for epidemiological modellers, and we thank the reviewer for encouraging us to highlight this more clearly. As described in our response to the editorial comments above, we have therefore followed the reviewer’s suggestion by adding a paragraph to the Results in which we summarise the methodological advance required to fit the models developed in our previous work to data from households rather than infector-infectee pairs (lines 137-144 and 150-154).

Weaknesses– The main weakness of the paper in its current form is that the analysis appears superficial, with a large amount of curve fitting and very little explanation. It would be beneficial if the authors delved more deeply into their results, especially with the mechanistic model. It would be very interesting to relate the changes in generation time to mechanisms of transmission.

While the primary aim of this research was to obtain updated generation time estimates and demonstrate the key principle that this important quantity is changing, in our revised submission we have extended the analyses within and around Figure 3 to delve deeper into the finding of a temporal decrease in the generation time.

First, we have added a new panel to Figure 3 (panel C in the revised submission) in which we show that the predicted decrease in generation time was accompanied by an increase in the proportion of pre-symptomatic transmissions, with a very high 83% of transmissions predicted to occur before symptom onset (among infectors who developed symptoms) in September-November (lines 325-332). We note in the Discussion (lines 570-572) that this finding is consistent with our hypothesis that a shorter generation time in the autumn months may have resulted from increased indoor contacts as the weather became colder, particularly among individuals without COVID-19 symptoms (whereas symptomatic hosts were still expected to self-isolate; lines 559-562 and 568-570).

Second, as suggested by the reviewer below, we have added a new figure (Figure 3—figure supplement 3) in which we compare the generation time distribution itself between the three different time periods (compared to Figure 3, where we focus on the mean and standard deviation of this distribution), as well as the distributions of the time from symptom onset to transmission (TOST) and the serial interval. Both models indicate that the transmission risk peaked earlier in infection for individuals infected in September-November compared to earlier months (lines 321-325).

Third, we have added a figure (Figure 3—figure supplement 5) in which we compare estimates of individual model parameters for the mechanistic model between the different time periods. As described in lines 348-354 of the revised manuscript, this showed that our finding of a shorter generation time and higher proportion of pre-symptomatic transmissions in September-November compared to earlier months may have resulted from any of: (i) an increase in the relative infectiousness of pre-symptomatic infectious infectors compared to symptomatic infectors (which is consistent with the hypothesis of increased indoor mixing among non-symptomatic individuals described above); (ii) a decrease in the (mean) duration of the symptomatic infectious period (which could, for example, result from faster isolation of symptomatic individuals); or (iii) a decrease in the (mean) time to infectiousness. However, since there was substantial overlap in the credible intervals for each individual parameter between the time periods, it was not possible to definitively identify the parameter(s) responsible for the observed change in the generation time (lines 354-357).

– The authors calculate the mean and standard deviation of the generation interval across three different time points; however, they only present one figure with the distribution of the generation time (Figure 2). It would be interesting to know how the generation time distribution changes in time, as opposed to just the mean and standard deviation. I believe that such an analysis would link nicely to their previous work, where they highlight the importance of ongoing public health measures such as contact tracing.

As described in our response to the previous point above, we have implemented this excellent suggestion in our revised submission (Figure 3—figure supplement 3 and lines 321-325).

I would like to congratulate the authors on a timely update to their work. I thoroughly enjoyed seeing their updated results, especially as some of the questions addressed have been of interest to myself. I do however have some recommendations.

We thank the reviewer for recognising the interest of updated estimates of the generation time and for their useful recommendations.

I understand that writing a rather mathematical paper for a general audience can be quite complicated, but I feel in this case that the authors have done themselves a disservice by not emphasising the technical concepts in the paper. At first read it appears that the authors have taken their model and fitted values, which is not particularly interesting. It was only once I made it to the Materials and methods section where I found the significant extension on previous work. I believe highlighting the adaptation of the likelihood function to account for the household level data was non–trivial and should be mentioned earlier (I believe this could be placed in the Results section), adding to the appeal of the paper. I note that susceptible depletion is mentioned in the main text, but I believe you should elaborate on how the likelihood function has been constructed to account for this.

We thank the reviewer for this helpful suggestion which has allowed us to improve the manuscript. As described above, we have followed the reviewer’s suggestion by describing earlier in the manuscript the methodological advance required to fit the models developed in our previous work to household data rather than data from infector-infectee pairs (lines 137-144 and 150-154). We agree that this adds to the appeal of this paper.

Throughout the work the posterior mean has been used as a point estimate for parameter values. I believe a more natural point estimate would be to choose the maximum of the posterior distribution. I notice that when looking at the posterior distributions of the mechanistic model (Figure S2), the maximum value of the posterior and the posterior mean differ by a wide mark for α_p and k_E/k_inc. The impact of this choice might be minimal, but I believe it should be investigated.

The mode of the joint posterior distribution of the fitted model parameters (i.e., the maximum a posteriori estimate) is not readily available as an output from the data augmentation MCMC approach that we used to fit the two models to the household data. Therefore, as in other studies using data augmentation MCMC (see, for example, the studies by Ferguson et al. (https://doi.org/10.1038/nature04017) and by Cauchemez et al. (https://doi.org/10.1002/sim.1912) to which we refer in our manuscript), we used the posterior mean as a point estimate. We state this justification for using the posterior mean in lines 162-166 of the revised manuscript.

A possible alternative is to obtain point estimates by estimating the mode of the marginal posterior distribution of each fitted parameter. As noted by the reviewer, this would have a substantial effect on point estimates of some fitted parameters for the mechanistic model. However, both methods of obtaining point parameter estimates lead to very similar inferred estimates of the generation time. For example, for the posterior parameter distributions for the mechanistic model shown in Figure 1—figure supplement 2 (the figure corresponding to Figure S2 in our initial submission), the inferred point estimate of the mean generation time is 5.9 days when using either posterior means or marginal posterior modes, and the point estimate of the standard deviation is 4.8 days in both cases.

It would be interesting to know how the generation time distribution changes in time, as opposed to just the mean and standard deviation. This would be a simple extension where they take the point estimates for multiple time points to show the temporal variation. I believe that such an analysis would link nicely to your previous work.

As noted above, we have implemented this excellent suggestion in our revised submission (Figure 3—figure supplement 3 and lines 321-325).

I am uncertain why the arguments of the paragraph at line 227 are required. It appears that the point is to justify the inclusion of a 1/n factor in the force of infection, however, I believe this is an obvious factor to include (I would use 1/(n–1) rather than 1/n though) that does not require parameter fitting to understand. If you were to consider a multigroup SIR model with varying population numbers the 1/(n–1), where n is the number of individuals in the group, is included so as the force of infection acts on the proportion of individuals that are susceptible. If this was not the case, then a different β would be required in each group. As you argue that the β value is a constant and does not vary between households it makes sense that the β value must be scaled by the number of individuals in the household, otherwise you would need a different β value for each house (which would be impossible to infer given the small household sizes).

This is an interesting point. We agree with the reviewer that frequency-dependent transmission is a natural assumption, and that 1/(n-1) may be a more natural choice of scaling factor than 1/n. We used the factor 1/n in most of our analyses since this is a common choice in the literature (see for example two papers by Cauchemez et al. (https://doi.org/10.1002/sim.1912 and https://doi.org/10.1371/journal.ppat.1004310) to which we refer in our manuscript). However, we also show in Figure 1—figure supplement 10 that the exact choice of either 1/n or 1/(n-1) has a minimal effect on our estimates of the generation time (see also lines 423-424 and 439-441 of the revised manuscript).

We felt it was important to confirm the robustness of our results to the assumption of frequency-dependent transmission because some previous studies have predicted household influenza transmission to be a somewhere between frequency- and density-dependent – for example, two studies by Ferguson et al. (https://doi.org/10.1038/nature04017) and Endo et al. (https://doi.org/10.1371/journal.pcbi.1007589) to which we refer in our manuscript predicted transmission to scale with 1/n^0.8 and 1/n^0.5, respectively. This motivation for including this sensitivity analysis in our work is now outlined in the Results of the revised manuscript (lines 408-414; this corresponds to the paragraph at line 227 in the original submission mentioned by the reviewer).

For reproducibility and transparency, I would like the authors to provide all code used to generate results, in line with eLife's policies on availability of data, software and research materials. This will allow other researchers to implement the methods they have developed on other data sets, but also enable confirmation that there is no coding mistakes.

We completely agree with the need to ensure that all code is publicly available. The code underlying our analyses is publicly available at https://github.com/will-s-hart/UK-generation-times. We include this link in the data availability section of our revised submission.

Reviewer #3:The authors have previously published a mechanistic model for inferring infectiousness profile that explicitly models dependence of the risk of onward transmission on the onset of symptoms on an individual. In the present study, they apply this model as well as another more commonly used model which assumes these two things (transmission risk and onset of symptoms) to be independent, to data from a household study conducted from March–Nov 2020 in the UK. Both the models find that the mean generation time in Sept–Nov 2020 is shorter than in the earlier periods of the study.This is well–presented study with careful analysis and extensive sensitive analysis which shows that the modelled estimates are robust to a range of assumptions.

We are pleased that the reviewer found our study to be well-presented and for recognising the significant sensitivity analyses that we performed to ensure that our results are robust.

[Editors’ note: what follows is the authors’ response to the second round of review.]

Essential revisions:1) While the observation of reduced generation times is both useful if true, and potentially plausible, it may not be robust. The overlap between the posterior estimates of generation times etc. are quite broad – and looking across three periods it doesn't seem like it would take much to change the trends in even the mean values.

This is an interesting point, which has motivated us to undertake an explicit quantitative comparison of the posterior estimates of the mean generation time between the different time periods. We found that the independent transmission and symptoms model indicated a 97% posterior probability of a shorter mean generation time in September-November 2020 compared to May-August, and the mechanistic model a 98% posterior probability. These comparisons are included in the Results of our revised submission (lines 273-282). These results provide quantitative evidence of the robustness of our finding of a shorter generation time in the autumn of 2020 compared to earlier months.

2) In particular, the size of the study is not that large, and in each household, it seems from the Miller paper, that only two PCR tests were taken – as the approach does not consider the impact of latent processes (i.e. missed infections) it would be important to know whether a slight bias in missed infections across periods would impact on the conclusions.

The reviewer is correct that two PCR tests were taken by each household member as part of the household study, but in fact antibody testing was also carried out (see for example lines 683-684 of the revised manuscript). We expect the combination of PCR and antibody testing to have minimised any possibility of missed infections.

We also conducted a sensitivity analysis (shown in Figure 1—figure supplement 12 and described in lines 425-433) in which we considered different assumptions regarding the infection status of 34 individuals for whom infection status could not be determined (these individuals did not return a positive PCR test and did not undertake antibody testing), obtaining almost identical estimates of the generation time under each assumption considered (although estimates of the overall infectiousness parameter, β0, were affected by the exact assumption). If a small number of infected individuals never returned a positive PCR test and tested negative for antibodies, then we similarly expect such potentially missed infections to have had a very small effect on our generation time estimates.

3) The authors also state (line 573) that "Potential bias towards more recent infection of the primary host if community prevalence is increasing, or less recent if prevalence is decreasing (Britton and Scalia 900 Tomba, 2019; Ferretti et al., 2020b; Lehtinen et al., 2021), was neglected." Could this also provide some possible explanation for the shift in generation times? Especially given that the justify their assumption in part on the analysis across individual months, and there are relatively few recruited households (on the order of 10 I think, based on Figure 3 in the supplement).

We have expanded our discussion of this important point in our revised submission (lines 541-556). In particular, we note the possibility that overrepresentation of shorter generation times in a growing epidemic may have contributed to our shorter estimated mean generation time for September-November 2020 (particularly in September and October, when national case numbers were mostly increasing) compared to earlier months of the study (when case numbers were mostly decreasing; lines 541-548). However, our mean generation time estimate for November 2020 (in which case numbers were mostly decreasing) is similar to the estimates for September and October 2020 (see Figure 3—figure supplement 4). This suggests that the effect of these background epidemic dynamics did not drive the temporal changes in the generation time that we inferred (lines 548-552). Finally, as mentioned by the reviewer, we note that an important caveat regarding this comparison between generation time estimates in individual months is the relatively small sample size per month (lines 552-553).

4) The authors also say that (line 150) " we corrected for the regularity of household contacts to derive more widely applicable estimates of the generation time. We did this by including a factor in the likelihood to account for each infected individual avoiding infection from household contacts that occurred prior to their actual time of infection (see Materials and Methods for full details of our approach)." This sounds really interesting and would greatly increase the generality of the outcome. But unfortunately, from the description in the material and methods I was not able to figure out exactly why this was – which doesn't mean it’s wrong of course, but it would be helpful to me to have a more detailed description.

We have expanded the relevant description in the Materials and Methods as requested (lines 848-860).

In our modelling approach, the instantaneous force of infection exerted by an infected host onto each susceptible individual in their household at time since infection τ is given byτCs+C1daD(t)dt=−aD(t)+bD(t)Cs+CI

The function *f*(τ) represents the (normalised) relative infectiousness profile of an infected host at each time since infection and is independent of the household structure. The total within-household force of infection on any susceptible individual, λ(*t*), essentially involves a sum of β(*t*) terms for each infected individual in the household. The probability of individual *k* becoming infected at time *t*_*k*_ requires both: (i) the individual to avoid infection before time *t*_*k*_; and (ii) the individual to then become infected at time *t*_*k*_.

In our previous *eLife* article (upon which this Research Advance builds), we considered transmission between known infector-infectee transmission pairs. In that analysis, we estimated *f*(τ) using a likelihood function that included a term corresponding to point (ii) above. However, as is common in studies estimating the generation time using data from infector-infectee pairs, we did not include a term corresponding to point (i) in that study – the exclusion of such a term amounts to an implicit assumption that contacts between the infector and infectee in each transmission pair are sporadic and of short duration, so that the probability of the infector transmitting the pathogen to the infectee before time τ is negligible (and similarly for the probability of the infectee being infected before time τ by an individual other than their eventual infector).

In this Research Advance, to estimate *f*(τ) from the household data, we added a term to the likelihood corresponding to point (i) – specifically, the factor exp(−Λ(*t*_*k*_)), where Λ(*t*) is the integral of λ(*t*) between times −∞ and *t*. This term represents the probability of avoiding infection from household contacts occurring before time *t*_*k*_. This probability may be non-negligible in the household context due to the high frequency of household contacts. Inclusion of this term therefore allowed us to correct for the regularity of household contacts to correctly derive estimates of *f*(τ) from the household data.

We also now clarify in the Discussion (lines 631-642) that the expected infectiousness profile, *f*(τ), provides a widely applicable estimate of the generation time distribution that is independent of the household size (specifically, *f*(τ) gives the generation time distribution under the assumption that a constant supply of susceptible individuals is available throughout the host’s course of infection). In principle, *f*(τ) can be used to calculate the generation time distribution of realised transmissions in different settings by combining this function with the contact network of those other settings – see for example the estimates of realised generation times in study households in Figure 1—figure supplement 4, which are shorter than our main generation time estimates in Figure 1 (which are derived from *f*(τ)) because of the regularity of household contacts and the depletion of susceptible individuals within households before longer generation times can be obtained.

5) The authors state that on line 163 that "point estimates for each model using the posterior means of fitted model parameters because the mode of the joint posterior distribution could not easily be calculated from the output of the MCMCprocedure." It would be important to know whether there are any correlations in the parameter posteriors that might make inappropriate.

As described in the sentence quoted by the reviewer (lines 161-165 of the revised manuscript), we used posterior means as point estimates of directly fitted model parameters. These point parameter values were then used to calculate point estimates of secondary quantities, such as the mean and standard deviation of the generation time distribution for the mechanistic model (please note that these two quantities were directly fitted for the independent transmission and symptoms model, but were secondary quantities for the mechanistic model), and the proportion of pre-symptomatic transmissions for both models.

We do not think that correlations between parameter posteriors would necessarily make this approach inappropriate. However, we do note here that an alternative method would be to first calculate the posterior distributions of secondary quantities using the output of the MCMC procedure (by calculating “current” estimates of these quantities at each step of the chain, as we did to obtain the violin plots shown in Figure 1), then calculate the means of these distributions. This method would account for correlations between the posterior distributions of fitted parameters, but we instead chose to use our approach as described above to ensure consistency of point estimates (for example, ensuring that if the generation time distribution in the independent transmission and symptoms model had the point estimate values of the mean and standard deviation, then the corresponding proportion of pre-symptomatic transmissions would also be given by the point estimate of that quantity). Nonetheless, we found the two approaches for calculating point estimates to give similar answers – for example, point estimates of the proportion of pre-symptomatic transmissions for the independent transmission and symptoms model using our method and the alternative approach were 0.72 and 0.72, respectively; point estimates of the proportion of pre-symptomatic transmissions for the mechanistic model were 0.74 and 0.73; point estimates of the mean generation time using the mechanistic model were 5.9 days and 6.0 days.

6) I spent some time trying to understand if there could be any issue causing the higher fraction of pre symptomatic transmission, which is the most unexpected result, and I could not find any obvious one. Same for the high variance of the generation time distribution (though this and the high pre–symptomatic transmission could be related). Hence, I think that these results can be published in the current form.

We are pleased the reviewer is happy for these results to be published in their current form.

On the other way, the temporal changes in generation time do not seem to account for the epidemic dynamics and therefore would be biased upward in Spring 2020 and downward in Autumn 2020 as observed. The authors are aware of that as they explain in the Discussion, but I think that the author should either correct for this effect in their approach or clarify better how this effect is accounted for and what may its contribution be.

As described in our response to point 3 above, we have expanded our discussion of the possibility that our generation time estimates were affected by background epidemic dynamics (lines 541-553 of the revised manuscript).

While methods exist for explicitly accounting for background epidemic dynamics in generation time estimates obtained using data from infector-infectee transmission pairs, we are not aware of such methods having been developed for estimating the generation time using household data. Therefore, we leave this interesting extension of our approach to future work (see lines 553-556).

Reviewer #1:The additional work done by the authors has been considerable and substantially increased the potential value of the work. In particular, the addition of data augmentation MCMC helps to provide greater depth to the outcomes, and the identification of declining generation times useful (especially if it could be established in 'real time' – i.e. rather than retrospectively, but to aid in understanding ongoing epidemics) and interesting.

We again thank the reviewer for their helpful comments on the earlier version of our manuscript, which helped us improve our work.

I do have a few concerns which in my view need to be addressed before it would be suitable for publication in eLife.First, while the observation of reduced generation times is both useful if true, and potentially plausible, it may not be robust. The overlap between the posterior estimates of generation times etc. are quite broad – and looking across three periods it doesn't seem like it would take much to change the trends in even the mean values.In particular, the size of the study is not that large, and in each household, it seems from the Miller paper, that only two PCR tests were taken – as the approach does not consider the impact of latent processes (i.e. missed infections) it would be important to know whether a slight bias in missed infections across periods would impact on the conclusions.The authors also state (line 573) that "Potential bias towards more recent infection of the primary host if community prevalence is increasing, or less recent if prevalence is decreasing (Britton and Scalia 900 Tomba, 2019; Ferretti et al., 2020b; Lehtinen et al., 2021), was neglected." Could this also provide some possible explanation for the shift in generation times? Especially given that the justify their assumption in part on the analysis across individual months, and there are relatively few recruited households (on the order of 10 I think, based on Figure 3 in the supplement).The authors also say that (line 150) " we corrected for the regularity of household contacts to derive more widely applicable estimates of the generation time. We did this by including a factor in the likelihood to account for each infected individual avoiding infection from household contacts that occurred prior to their actual time of infection (see Materials and Methods for full details of our approach)." This sounds really interesting and would greatly increase the generality of the outcome. But unfortunately, from the description in the material and methods I was not able to figure out exactly why this was – which doesn't mean it’s wrong of course, but it would be helpful to me to have a more detailed description.The authors state that on line 163 that "point estimates for each model using the posterior means of fitted model parameters because the mode of the joint posterior distribution could not easily be calculated from the output of the MCMC procedure." It would be important to know whether there are any correlations in the parameter posteriors that might make inappropriate.

We thank the reviewer for these additional comments, which we address under Essential Revisions above.

Reviewer #4:This excellent paper suggests that despite extensive studies, we have not yet reached a full understanding of the generation time of SARS–CoV–2. The study is a robust examination of the subject of generation time within households in UK, which may not be representative of transmission in other contexts. It is unclear to the reviewers if temporal changes in generation time are real and attributable to e.g. the appearance of B.1.177.This work is sound. While surprising, the results are supported by multiple statistical/modelling approaches and robustness analyses, and believable.

We thank the reviewer for their helpful comments. The suggestion that the emergence of the B.1.177 lineage may have contributed to our finding of a temporal decrease in the generation time is interesting, and we mention this possibility in the Discussion of our revised manuscript (lines 522-527).

The three most striking results are:1) The width of the generation time distribution is much wider than in previous works. While this is undoubtedly surprising, the explanation by the authors is believable: home quarantine in the UK is probably less effective in stopping late transmissions within households and may even amplify them.

We are pleased that the reviewer agrees with this hypothesis for the cause of the relatively high reported standard deviation of the generation time distribution.

2) The fraction of pre-symptomatic transmissions is >70%, quite high compared to most previous estimates. Combined with the high number of fully asymptomatic individuals, it would imply that <20% of transmissions come from individuals showing symptoms. This result seems also hard to square with the previous one, which would suggest a wide distribution of TOST. Of course, this estimate may be affected by the setting, since the analysis is restricted to households and therefore a higher force of infection.

We agree with the reviewer that our estimates for the proportion of pre-symptomatic transmissions are high compared with some previous estimates, although similar or higher estimates do exist elsewhere in the literature (including in our previous paper in *eLife*, which this Research Advance builds on), as described in lines 191-198 of our manuscript.

In fact, the TOST distribution for the independent transmission and symptoms model shown in Figure 2B has a higher standard deviation (5.8 days) than the corresponding generation time distribution in Figure 2A (4.9 days). For the mechanistic model, the generation time and TOST distributions have similar standard deviations (4.8 days and 4.9 days, respectively). These standard deviations are reported in Appendix 1-table 4; to highlight this, we have added a reference to this table to the main text of our revised submission (lines 242-243). Therefore, the reviewer is correct in expecting our generation time distribution estimates to correspond to relatively wide TOST distributions, but the proportion of pre-symptomatic transmissions is nonetheless high in both models.

3) According to this work, the generation time changed between spring 2020 and autumn 2020 in the UK. This corresponds to the arrival of the B.1.177 lineage, probably more infectious than previous variants, but also to a different epidemiological phase of the epidemic: lockdown followed by gradual reopening in spring/summer, with a corresponding decrease in incidence, then a new wave in autumn with an increase in the number of cases until November. The authors do not correct for this epidemiological dynamic, therefore leaving open the possibility that it would cause an apparent change in generation time similar to the observed one. Other explanations (e.g. behavioural or reporting ones) may be possible.

As described above, in our revised submission we discuss the possibility that the arrival of the B.1.177 lineage (lines 522-527) and/or background epidemiological dynamics (lines 541-553) may have contributed to our finding of a temporal change in the generation time.

It is important to remark that many of the results of the mechanistic model may be affected by the assumption that longer incubation intervals correspond to higher infectiousness. The agreement with the results of the simpler model with independent incubation period and generation time implies that this assumption is not relevant for the main results (with the possible exception of the longer mean generation time).

We contend that it is realistic to assume (as is the case in the mechanistic model) that individuals with longer incubation periods will (on average) have longer pre-symptomatic infectious periods, and therefore generate more transmissions, compared to those with shorter incubation periods. However, we agree that this assumption is likely to affect estimates of epidemiological quantities using that model (but does not affect our main finding of a temporal decrease in the generation time). We therefore now note this assumption when first describing the mechanistic model in the Introduction of our revised submission (lines 111-113).

Recommendations:The results of the paper look really robust.I spent some time trying to understand if there could be any issue causing the higher fraction of pre symptomatic transmission, which is the most unexpected result, and I could not find any obvious one. Same for the high variance of the generation time distribution (though this and the high pre–symptomatic transmission could be related). Hence, I think that these results can be published in the current form.

We are pleased that our results look robust to the reviewer, and that they believe our results can be published in the current form.

On the other way, the temporal changes in generation time do not seem to account for the epidemic dynamics and therefore would be biased upward in Spring 2020 and downward in Autumn 2020 as observed. The authors are aware of that as they explain in the Discussion, but I think that the author should either correct for this effect in their approach or clarify better how this effect is accounted for and what may its contribution be.

Please see the Essential Revisions above for our response to this comment. We again thank the reviewer for their helpful comments and suggestions.